# Tip cell-specific requirement for an atypical Gpr124- and Reck-dependent Wnt/β-catenin pathway during brain angiogenesis

Benoit Vanhollebeke[1,2,3]\*, Oliver A Stone[1,2], Naguissa Bostaille[3], Chris Cho[4], Yulian Zhou[4], Emilie Maquet[1], Anne Gauquier[3], Pauline Cabochette[3], Shigetomo Fukuhara[5], Naoki Mochizuki[5], Jeremy Nathans[4,6,7], Didier YR Stainier[1,2]

[1]Department of Biochemistry and Biophysics, University of California, San Francisco, San Francisco, United States; [2]Department of Developmental Genetics, Max Planck Institute for Heart and Lung Research, Bad Nauheim, Germany; [3]Institut de Biologie et de Médecine Moléculaires, Université Libre de Bruxelles, Gosselies, Belgium; [4]Department of Molecular Biology and Genetics, Johns Hopkins University School of Medicine, Baltimore, United States; [5]Department of Cell Biology, National Cerebral and Cardiovascular Center Research Institute, Osaka, Japan; [6]Department of Neuroscience, Howard Hughes Medical Institute, Johns Hopkins University School of Medicine, Baltimore, United States; [7]Department of Ophthalmology, Johns Hopkins University School of Medicine, Baltimore, United States

\*For correspondence: Benoit. Vanhollebeke@ulb.ac.be

**Abstract** Despite the critical role of endothelial Wnt/β-catenin signaling during central nervous system (CNS) vascularization, how endothelial cells sense and respond to specific Wnt ligands and what aspects of the multistep process of intra-cerebral blood vessel morphogenesis are controlled by these angiogenic signals remain poorly understood. We addressed these questions at single-cell resolution in zebrafish embryos. We identify the GPI-anchored MMP inhibitor Reck and the adhesion GPCR Gpr124 as integral components of a Wnt7a/Wnt7b-specific signaling complex required for brain angiogenesis and dorsal root ganglia neurogenesis. We further show that this atypical Wnt/β-catenin signaling pathway selectively controls endothelial tip cell function and hence, that mosaic restoration of single wild-type tip cells in Wnt/β-catenin-deficient perineural vessels is sufficient to initiate the formation of CNS vessels. Our results identify molecular determinants of ligand specificity of Wnt/β-catenin signaling and provide evidence for organ-specific control of vascular invasion through tight modulation of tip cell function.

## Introduction

Endothelial cells (ECs) acquire organ-specific characteristics to adapt to the requirements of their host tissues. The central nervous system (CNS) vascular microenvironment serves as a paradigm for blood vessel specialization because CNS ECs develop a set of junctional, cellular trafficking, and metabolic properties, collectively called the blood–brain barrier (BBB), that protect the CNS from blood-borne toxins and pathogens. Brain angiogenesis and barriergenesis are temporally coupled through a distinct, and tissue-specific, developmental program (*Obermeier et al., 2013*; *Engelhardt and Liebner, 2014*; *Vallon et al., 2014*). The best characterized class of angiogenic and BBB-inductive signals operates through the Wnt/β-catenin pathway (canonical Wnt signaling) (*Xu et al., 2004*; *Liebner et al., 2008*; *Stenman et al., 2008*; *Daneman et al., 2009*; *Ye et al., 2009*), with distinct sets

**eLife digest** Organs develop alongside the network of blood vessels that supply them with oxygen and nutrients. One way that new blood vessels grow is by sprouting out of the side of an existing vessel, via a process called angiogenesis. This process relies on signals that are received by the endothelial cells that line the inner wall of blood vessels, and that direct the cells to form a new 'sprout', consisting of tip and stalk cells.

In the developing brain, the Wnt/β-catenin signaling pathway helps direct the formation of blood vessels. In this pathway, a member of a protein family called Wnt signals to specific proteins on the surface of the cells lining the blood vessels. Much effort has gone into uncovering the identity of these proteins, with many studies looking at blood vessel development in the brain of mouse embryos.

In this study, Vanhollebeke et al. turned to zebrafish embryos to uncover new regulators of angiogenesis and define their roles during the multi-step process of blood vessel development in the brain. A variety of experimental techniques were used to alter and study the activity of different Wnt signaling pathway components. These experiments revealed that the Wnt7a and Wnt7b proteins signal to an endothelial cell membrane protein complex containing the proteins Gpr124 and Reck.

Vanhollebeke et al. then created 'mosaic' zebrafish embryos, which contained two genetically distinct types of cells—cells that were missing one of the components of Wnt/β-catenin signaling pathway, and wild-type cells. Visualizing the growth of the vessels showed that all the new blood vessels that sprouted had normal tip cells. However, the cells in the stalk of the sprout could be either normal or missing a signaling protein.

These findings demonstrate that Wnt/β-catenin signaling controls the pattern of blood vessel development in the brain by acting specifically on the invasive behaviors of the tip cells of new sprouts, a cellular mechanism that allows efficient organ-specific control of vascularization.

of ligands, receptors, and co-receptors controlling vascular development in different CNS locations. For example, in the embryonic forebrain and ventral spinal cord, neural progenitor-derived Wnt7a and Wnt7b activate Wnt/β-catenin signaling in ECs to control both angiogenesis and BBB formation (*Liebner et al., 2008*; *Stenman et al., 2008*; *Daneman et al., 2009*). In the retina, the Muller glia-derived ligand Norrin, in conjunction with the receptor Frizzled4 (Fz4), co-receptor Lrp5, and co-activator Tspan12, mediate Wnt/β-catenin signaling to control angiogenesis and blood-retina barrier (BRB) formation and maintenance (*Xu et al., 2004*; *Junge et al., 2009*; *Ye et al., 2009*).

In addition to classical components of the Wnt/β-catenin pathway, such as Frizzled receptors and Lrp5/Lrp6 co-receptors (*Zhou et al., 2014*), recent evidence indicates that a unique signal transduction complex containing Gpr124, an orphan receptor of the adhesion GPCR family, operates specifically in CNS ECs to promote Wnt7a and Wnt7b angiogenic signaling (*Kuhnert et al., 2010*; *Anderson et al., 2011*; *Cullen et al., 2011*; *Zhou and Nathans, 2014*; *Posokhova et al., 2015*). In mouse embryos, eliminating neuroepithelial Wnt7a and Wnt7b, or endothelial Gpr124 or β-catenin, leads to reduced CNS angiogenesis with production of abnormal vascular structures, termed glomeruloids, that fail to acquire BBB characteristics (*Liebner et al., 2008*; *Stenman et al., 2008*; *Daneman et al., 2009*; *Kuhnert et al., 2010*; *Anderson et al., 2011*; *Cullen et al., 2011*). Whether these vascular malformations are exclusively the result of defective endothelial Wnt/β-catenin signaling or are also influenced by signals from hypoxic tissues has not been determined (*Sundberg et al., 2001*; *Cullen et al., 2011*).

At present, the cellular mechanisms by which Wnt/β-catenin signaling controls the complex and multistep process of CNS blood vessel formation remain largely unexplored. Here, we leverage the zebrafish model to study the molecular machinery governing Wnt-dependent brain angiogenesis using a combination of targeted mutagenesis, morpholino knock-downs, RNA injections, genetic mosaics, and single-cell resolution real-time imaging. We find that EC-specific and Gpr124-dependent Wnt/β-catenin signaling is required for angiogenic sprouting throughout the zebrafish brain. In addition, we identify Reck (reversion-inducing-cysteine-rich protein with Kazal motifs), a GPI-anchored MMP inhibitor and angiogenic modulator (*Oh et al., 2001*), as a novel and essential activator of Wnt/β-catenin signaling during CNS angiogenesis, and we show that Gpr124 and Reck physically

interact and strongly synergize in mammalian cells to promote Wnt/β-catenin signaling exclusively via Wnt7a and Wnt7b. Finally, by using live imaging of genetically mosaic animals, we have discovered a tip cell-autonomous requirement for Gpr124- and Reck-dependent Wnt/β-catenin signaling during sprouting angiogenesis in the CNS. These experiments demonstrate that Wnt/β-catenin signaling specifically regulates tip cell function and reveal that coordination of tip and stalk cell behaviors within nascent vessels and organ-specific specialization, generally viewed as distinct aspects of vascular development, can in fact be tightly coupled.

## Results

### *gpr124* mutants lack CNS blood vessels and dorsal root ganglia sensory neurons

To examine the function of Gpr124 during zebrafish development, we generated two *gpr124* mutant alleles using TAL effector nucleases (*Cermak et al., 2011*; *Dahlem et al., 2012*). The TALEN pairs were directed towards sequences within exons 7 and 16, corresponding to the Ig-like domain and second transmembrane helix, respectively (*Figure 1A*). We identified frame-shift mutant alleles, *gpr124$^{s984}$* and *gpr124$^{s985}$*, which lead to premature stop codons after 10 and 39 amino acid-long missense segments following the lesion site (*Figure 1—figure supplement 1*). Heterozygous carriers of either mutant allele display no obvious anatomical or behavioral phenotype. Homozygous *gpr124$^{s984}$* and *gpr124$^{s985}$* mutants, although morphologically indistinguishable from wild-type siblings (*Figure 1B*), exhibit specific and highly penetrant brain vascular defects (*Figure 1C, D*). The initial assembly of the perineural vessels is unaffected in the absence of Gpr124. Between 28 and 32 hpf (hours post fertilization), the paired ventro-lateral primordial hindbrain channels (PHBC) and primordial midbrain channels (PMBC) that extend along the rostro–caudal axis, establish wild-type-like connections with the medial basilar artery (BA) and the more rostral V-shaped posterior communicating segments (PCS).

From 32 hpf in wild-type embryos, the intracerebral central arteries (CtAs) begin to form by angiogenic sprouting from the dorsal wall of the PHBCs. These sprouts progress dorso-medially into the neural tissues to connect with the basilar artery and by 36 hpf, an average of four sprouts per hindbrain hemisphere have formed (*Bussmann et al., 2011*; *Fujita et al., 2011*; *Ulrich et al., 2011*). In contrast, *gpr124* mutants completely lack these forming vessels (*Figure 1C*). Defects in CNS vascularization are fully penetrant in *gpr124* mutants and at 60 hpf the entire brain remains avascular, while intersegmental vessel (ISV) sprouting from the dorsal aorta appears to be unaffected (*Figure 1C, D, E*, *Videos 1 and 2*). Injection of an anti-sense morpholino targeting the splice donor site of *gpr124* exon 6 dose-dependently mimicked the mutant vascular phenotype (*Figure 1E*) and both the mutant and morphant phenotypes could be partially rescued by the injection of RNA encoding the full-length receptor (*Figure 1F*). Asymmetric or unilateral rescues are sometimes observed (*Figure 2G*), possibly as a result of uneven distribution of the injected RNA within the yolk cell. Gpr125, a closely related adhesion GPCR, did not demonstrate angiogenic potential in a similar rescue assay. We mapped the functional differences between the receptors to components of the extracellular domain (*Figure 1F*).

Remarkably, *gpr124* mutant zebrafish can proceed through organogenesis in the complete absence of intracerebral blood vessels and approximately half of them reach adulthood (*Figure 1—figure supplement 2*), some becoming fertile. Adult *gpr124* mutants exhibit a CNS vascular network that appears of equal density to that of wild-type animals (*Figure 1G*, upper panels), indicating that CNS vascularization can ultimately occur in the absence of Gpr124. This late-onset brain vascularization program starts after 5 dpf (days post fertilization) with a varying degree of expressivity (*Figure 1—figure supplement 2*). The adult *gpr124*-deficient CNS vessels can acquire all tested BBB characteristics as revealed by immunostaining for Slc2a1 (Glut1), Pgp and by permeability assays (*Figure 1G*).

Non-transgenic homozygous *gpr124* mutants are first distinguishable from wild-type siblings by their reduced growth rate and eventually by the disrupted pigmentation patterns of their skin after metamorphosis (after 14 dpf). In contrast to the regular pattern of alternating longitudinal melanophore-rich dark stripes and light interstripes found in wild-types, *gpr124* mutants exhibit discontinuous stripes formed by clusters of melanophores bordered by xanthophores and iridophores, most prominently in the dorsal aspect of the trunk (*Figure 2A*). Metamorphic

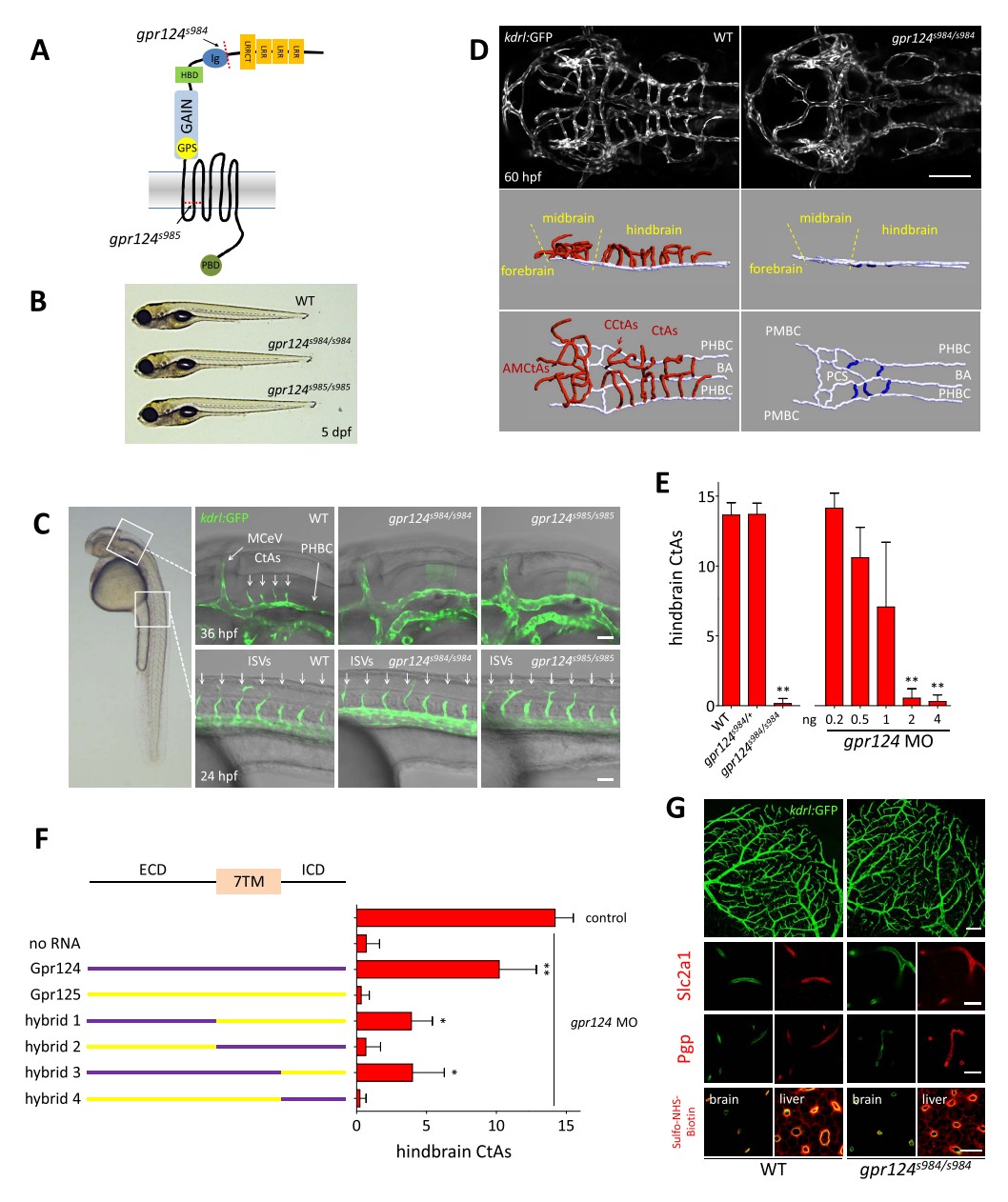

**Figure 1**. CNS vascular defects in *gpr124* mutants. (**A**) Schematic representation of Gpr124 structure and TALEN target site locations corresponding to *gpr124^{s984}* and *gpr124^{s985}* alleles. LRR: leucine-rich repeats; LRRCT: leucine-rich repeat C-terminal domain; Ig: Ig-like domain; HBD: hormone binding domain; GAIN: GPCR-autoproteolysis inducing domain; GPS: GPCR proteolysis site; PBD: PDZ binding domain. (**B**) Lateral views of wild-type, *gpr124^{s984/s984}* and *gpr124^{s985/s985}* larvae at 5 dpf. (**C**) Lateral views of wild-type, *gpr124^{s984/s984}* and *gpr124^{s985/s985}* *Tg(kdrl:GFP)* embryos at 36 hpf (hindbrain region, upper panels) and 24 hpf (trunk region, bottom panels). MCeV: middle cerebral vein. Scale bar, 50 μm. (**D**) Maximal intensity projection of a confocal *z*-stack of the cranial vasculature of *Tg(kdrl:GFP)* wild-type and *gpr124^{s984/s984}* embryos at 60 hpf in dorsal views (anterior to the left) and wire diagram of the brain vasculature in lateral (middle panels) and dorso-lateral (bottom panels) views. Red vessels in the 3D renderings represent the intra-cerebral central arteries (CtAs), blue vessels represent the extra-cerebral connections between the PHBC and BA lining the hindbrain ventrally, and gray vessels represent the perineural vessels (PHBC, PMBC, BA, and PCS) to which the central arteries connect in wild-type embryos. Scale bar, 100 μm. (**E**) Quantification of hindbrain CtAs upon Gpr124 depletion in 60 hpf embryos. (**F**) Quantification of hindbrain CtAs in control and *gpr124* morphants at 60 hpf after injection at the one-cell stage of 100 pg RNA encoding the depicted receptors or Gpr124/Gpr125 hybrid receptors. (**G**) Vasculature of wild-type and *gpr124* mutant adults. Single plane confocal

*Figure 1. continued on next page*

*Figure 1. Continued*

image of the vascular network (upper panels: scale bar, 100 µm) and immunostaining for Slc2a1 and Pgp in sections through the optic tectum (middle panels: scale bars, 20 µm). Evaluation of the optic tectum and liver vessel permeability by fluorescent streptavidin labelling (red signal) 60 min after intracardial injection of sulfo-NHS-biotin in live animals (bottom panels; scale bar, 20 µm). In all panels, values represent means ± SD (*p < 0.05; **p < 0.01; Kruskal–Wallis test). Morpholino and RNA injections were performed as described in 'Methods'.

The following figure supplements are available for figure 1:

**Figure supplement 1**. Generation of *gpr124* mutant zebrafish.

**Figure supplement 2**. *gpr124* mutant survival curve and vascular phenotypes at 5 and 14 dpf.

---

melanophores derive from a postembryonic stem cell population residing in close association with the segmentally arranged dorsal root ganglia (DRG) (*Dooley et al., 2013*), and adult pigmentation defects have been previously correlated with abnormal DRG formation (*Budi et al., 2008*; *Honjo et al., 2008*; *Malmquist et al., 2013*). Accordingly, *gpr124* mutants fail to form DRGs with both *sox10*:mRFP+ satellite glial cells and *ngn1*:EGFP+ neurons missing in most segments at 72 hpf (*Figure 2B, C*). On rare occasions, ganglia could be identified in the anterior-most segments of the trunk. The DRG defects do not result from the initial failure to specify the neuroglial lineage, as *sox10*:mRFP+ cells formed transient aggregates in all segments of *gpr124* mutants at earlier stages. The ganglia however never contained *ngn1*:EGFP+ neurons as seen in wild-type siblings and were not retained at later stages (*Figure 2D*). Other neural crest derivatives, like the embryonic pigment cells and lateral line glia, appear to develop normally, ruling out a general requirement for Gpr124 in neural crest-derived tissues.

## Reck is required for brain vascular development

In order to identify the components of the molecular pathway through which Gpr124 operates, we tested whether previously described regulators of DRG formation (*Budi et al., 2008*; *Honjo et al., 2008*; *Prendergast et al., 2012*; *Malmquist et al., 2013*) could, like Gpr124, additionally control CNS angiogenesis. Using a morpholino knock-down approach, we identified Reck, a GPI-anchored MMP inhibitor and angiogenic modulator (*Oh et al., 2001*) as a novel essential regulator of brain vascularization (*Figure 2E*). Reck and Gpr124 knockdown embryos exhibit identical CNS-specific vascular defects without detectable ISV or gross morphological phenotypes (*Figure 2F, I*). Other DRG

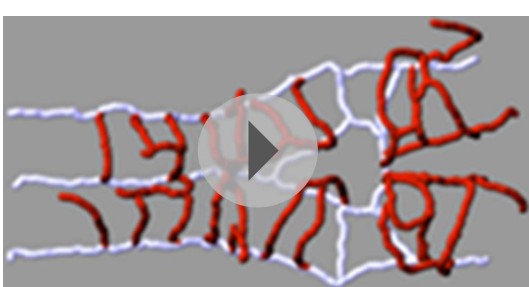

**Video 1.** Wild-type cerebral vasculature at 60 hpf. Three-dimensional rotation of a wire diagram representation (Imaris; Bitplane) of the wild-type zebrafish cerebral vasculature at 60 hpf. Red vessels in the 3D renderings represent the intra-cerebral central arteries (CtAs), and gray vessels represent the perineural vessels (PHBC, PMBC, BA, and PCS) to which the central arteries connect. Scale bar, 100 µm.

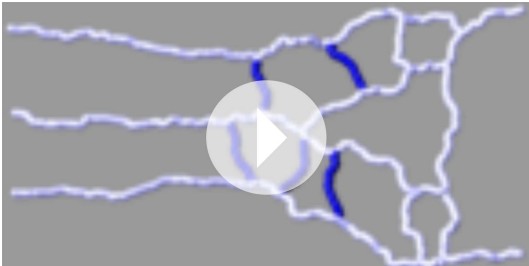

**Video 2.** *gpr124*s984/s984 cerebral vasculature at 60 hpf. Three-dimensional rotation of a wire diagram representation (Imaris; Bitplane) of the *gpr124*s984/s984 cerebral vasculature at 60 hpf. Blue vessels represent the extra-cerebral connections between the PHBC and BA lining the hindbrain ventrally, and gray vessels represent the perineural vessels (PHBC, PMBC, BA, and PCS). Scale bar, 100 µm.

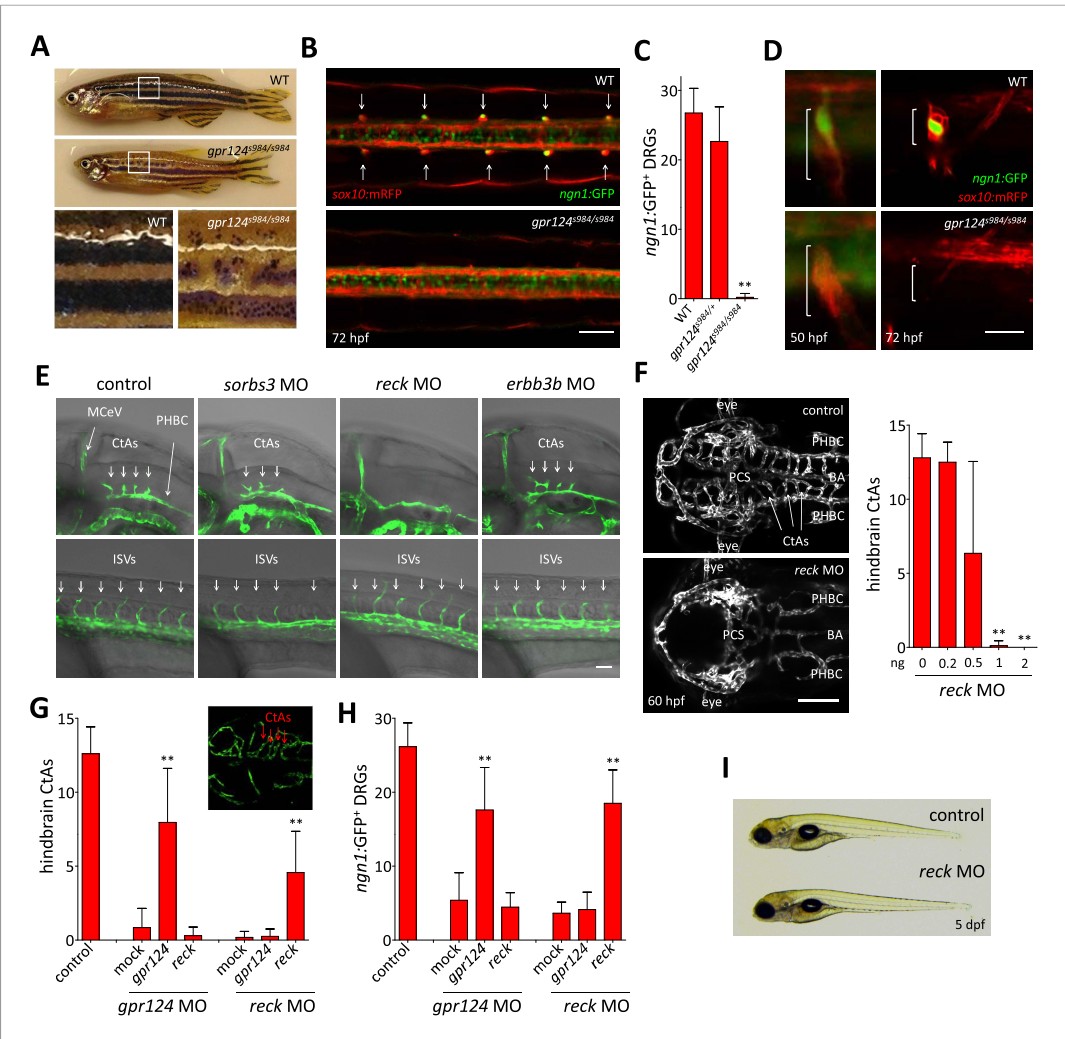

**Figure 2**. Essential regulation of CNS angiogenesis by Reck. (**A**) Lateral views of skin pigmentation patterns of wild-type and *gpr124^{s984/s984}* adults. Bottom panels are high magnification images of the upper panel boxed areas. (**B**) Dorsal views of DRGs in the trunk region of wild-type and *gpr124^{s984/s984}* Tg(ngn1:GFP); Tg(sox10:mRPF) larvae at 72 hpf; arrows point to DRGs (anterior to the left). Scale bar, 50 μm. (**C**) Quantification of *ngn1*:GFP+ DRGs in 72 hpf wild-type, *gpr124^{s984/+}* and *gpr124^{s984/s984}* larvae. *ngn1*:GFP+ DRGs were counted on one side of the larvae. (**D**) Lateral views of DRG in *Tg(ngn1:GFP);Tg(sox10:mRPF)* wild-type and *gpr124^{s984/s984}* animals. Scale bar, 20 μm. (**E**) Lateral views (anterior to the left) of *Tg(kdrl:GFP)* control and morphant embryos at 36 hpf (hindbrain region, upper panels) and 24 hpf (trunk region, bottom panels). Arrows point to the forming CtAs (upper panels) and ISVs (lower panels). Scale bar, 50 μm. (**F**) Maximal intensity projection of a confocal z-stack of *Tg(kdrl:GFP)* wild-type and *reck* morphant cranial vasculature at 60 hpf in dorsal views (anterior to the left; scale bar, 100 μm) and quantification of hindbrain CtAs after Reck downregulation by anti-sense morpholino injections at various doses. (**G**) Quantification of hindbrain CtAs in control and *gpr124* or *reck* morphants at 60 hpf after injection at the one-cell stage of 100 pg RNA encoding Gpr124 or Reck. The insert shows a typical rescue. (**H**) Quantification of *ngn1*:GFP+ DRGs in control and *gpr124* or *reck* morphants at 72 hpf after injection at the one-cell stage of 200 pg RNA encoding Gpr124 or Reck. *ngn1*:GFP+ DRGs were counted on one side of the larvae. (**I**) Lateral view of wild-type and *reck* morphants at 5 dpf. In all panels, values represent means ± SD (*p < 0.05; **p < 0.01; Kruskal–Wallis test). Morpholino and RNA injections were performed as described in 'Methods'.

regulators like Sorbs3 (*Malmquist et al., 2013*) and Erbb3b (*Honjo et al., 2008*) do not appear to modulate CNS angiogenesis specifically (*Figure 2E*). Reck knock-down dose-dependently blocked brain vascularization (*Figure 2F*), and at doses of morpholino above 1 ng, brains remained avascular. As previously reported, *reck* morphants completely lack DRGs at 72 hpf (*Prendergast et al., 2012*)

despite normal initial specification of sox10:mRFP⁺ neuroglial cells, much like gpr124 mutants (*Figures 2D and 3C*).

The remarkable phenotypic similarities observed after gpr124 and reck knock-downs in two settings of distinct embryological origin led us to probe their functional relationship. In both the CNS vascular and peripheral neurogenic settings, ectopic expression of Gpr124 or Reck could compensate for their respective loss-of-function but no functional epistatic relationship could be detected, that is, the absence of one protein could not be rescued by overexpression of the other (*Figure 2G, H*). This result is compatible with one of the following two scenarios: Gpr124 and Reck act in independent parallel pathways, or they act in concert to control a common signaling pathway during both CNS angiogenesis and DRG neurogenesis.

## Gpr124 and Reck control Wnt/β-catenin signaling

Wnt/β-catenin signaling controls CNS vascular formation in mouse (*Liebner et al., 2008*; *Stenman et al., 2008*; *Daneman et al., 2009*) and instructs sensory neural cell fates during DRG development (*Hari et al., 2002*; *Lee et al., 2004*). We tested whether Gpr124 or Reck participate in Wnt/β-catenin signaling during these processes, by evaluating the expression of Wnt/β-catenin reporter transgenes. We could detect *Tg(7xTCF-Xla.Siam:GFP)* expression in a subset of wild-type ECs of the PHBCs starting at 26 hpf. Gradually, both the intensity and the number of GFP⁺ cells increased and these cells contributed to the PHBC, BA, and CtAs of 48 hpf wild-type embryos. In contrast, no GFP⁺ ECs could be detected in the PHBC or BA after gpr124 or reck knock-down (*Figure 3A*), while the expression of the *7xTCF-Xla.Siam:GFP* transgene appeared unaffected in the surrounding neural tissues in the absence of Gpr124 or Reck. Examination of an endothelial-specific Wnt/β-catenin reporter transgene, in which the fusion of the DNA binding domain of Gal4 to the β-catenin binding domain of TCF4 acts as a sensor for nuclear β-catenin (*Kashiwada et al., 2015*), confirmed that Gpr124 and Reck are both required for Wnt/β-catenin signaling in the PHBCs (*Figure 3B*). Similarly, both proteins are required to establish *Tg(7xTCF-Xla.Siam:GFP)* expression in the forming DRGs at 54 hpf (*Figure 3C*). We next tested whether Wnt/β-catenin signaling inhibition could mimic the phenotypes induced by Gpr124 or Reck depletion. Global heat-shock induced expression of Dickkopf1 (*Stoick-Cooper et al., 2007*), the secreted inhibitor of Wnt/β-catenin signaling or Axin1 (*Kagermeier-Schenk et al., 2011*), a component of the GSK-3/Axin/APC β-catenin destruction complex, induced respectively a partial and near-complete inhibition of brain vascular development. Similarly, pharmacological inhibition of Wnt/β-catenin signaling by the Axin-stabilizing compound IWR-I dose dependently blocked brain vessel formation (*Figure 3D*). DRG neurogenesis was similarly sensitive to Axin1 overexpression (*Figure 3E*). These observations suggest that the downregulation of Wnt/β-catenin signaling in Gpr124 or Reck deficient embryos could explain their angiogenic and neurogenic defects. We tested this hypothesis by artificially restoring β-catenin levels through GSK-3β inhibition using structurally distinct compounds. These drugs could partially restore brain angiogenesis (*Figure 3F*) and completely restore DRG neurogenesis (*Figure 3G*) in gpr124 morphants.

## Reck and Gpr124 synergize to co-activate Wnt7a/Wnt7b signaling in reporter cells

In a previous study, we found that mouse Gpr124 co-activates Wnt/β-catenin signaling via Wnt7a/Wnt7b in a reporter cell line (Super Top Flash; STF [*Xu et al., 2004*]) and that signaling was further enhanced by co-transfection with Fz4 and Lrp5 (*Zhou and Nathans, 2014*). (We note that RNAseq analysis of STF cells showed low level expression of many Wnt signaling components, including Frizzled receptors, Lrp co-receptors, Gpr124, and Reck [*Zhou and Nathans, 2014*], which likely accounts for the signals observed when individual components are omitted from the transfection.) To determine whether Reck influences the Gpr124 dependence of Wnt7a- and Wnt7b-induced signaling, we co-transfected various combinations of Wnt7a or Wnt7b, Fz4, Lrp5 or Lrp6, Gpr124, and Reck into STF cells (*Figure 4A*). These experiments showed that Reck dramatically synergizes with Gpr124 in activating Wnt/β-catenin signaling in response to Wnt7a and Wnt7b, and that signaling is further increased by co-transfection with Fz4 together with Lrp5 or Lrp6. Both Gpr124 and Reck show well behaved dose–response curves, with synergistic activity over a wide range of DNA concentrations (*Figure 4B*).

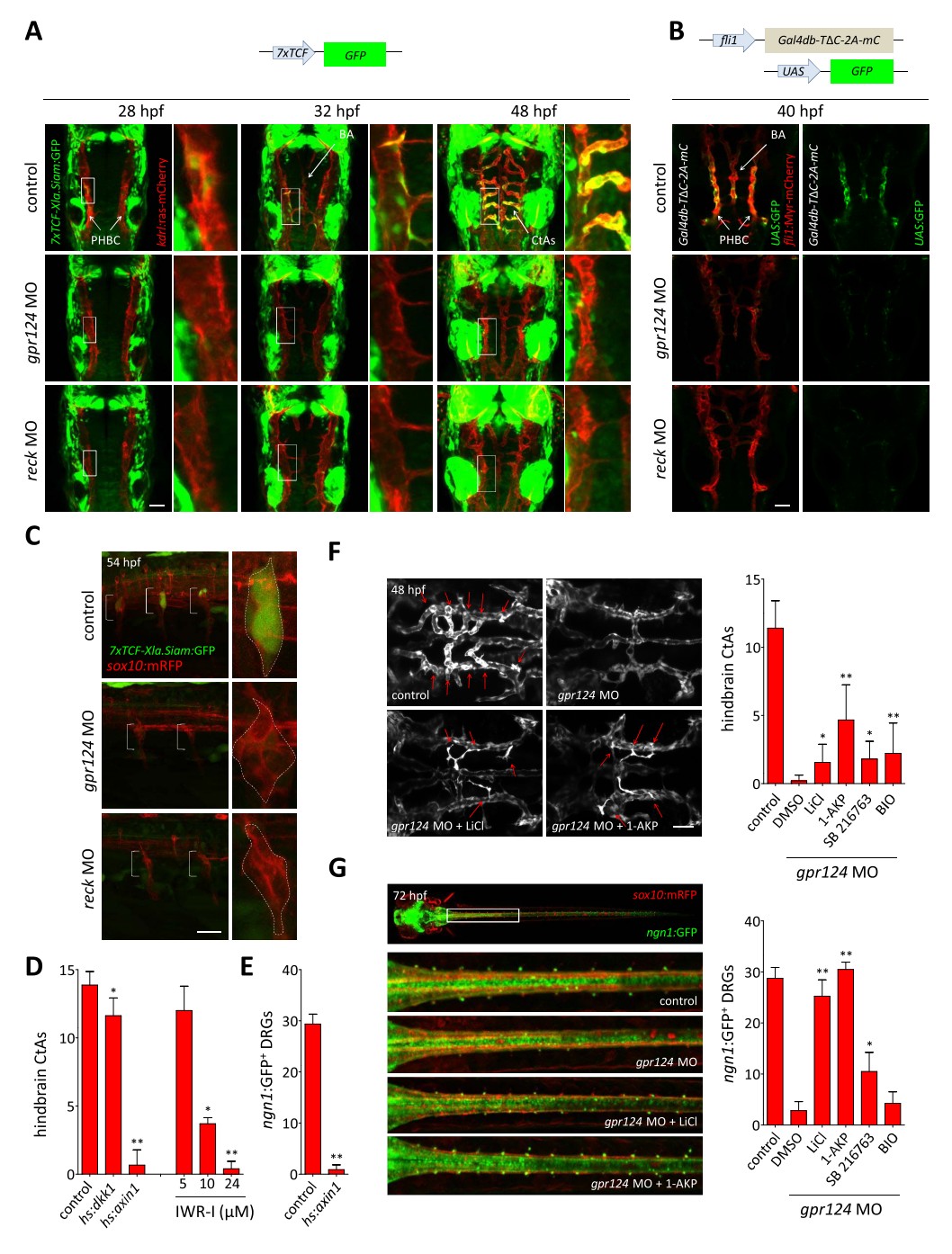

**Figure 3**. Wnt/β-catenin signaling is controlled by Gpr124 and Reck. (**A**) Maximal intensity projection of a confocal z-stack of *Tg(7xTCF-Xla.Siam*:GFP*)* Wnt/β-catenin reporter expression during brain vascular development in wild-type, *gpr124* or *reck* morphant *Tg(kdrl:ras-mCherry)* embryos. (**B**) Activity of the endothelial-specific Wnt/β-catenin reporter in the PHBC and BA in wild-type, *gpr124* or *reck* morphant *Tg(fli1:Myr-mcherry)* embryos at 40 hpf. (**C**) Lateral view of *Tg(7xTCF-Xla.Siam*:GFP*)* Wnt/β-catenin reporter expression at 54 hpf in the trunk region of wild-type, *gpr124* or *reck* morphant *Tg(sox10:mRPF)* embryos (anterior to the left). (**D**) Quantification of hindbrain CtAs at 60 hpf after genetic and pharmacological inhibition of Wnt/β-catenin signaling. Heat-shock and pharmacological inhibition were performed as described in 'Methods'. (**E**) Quantification of *ngn1*:GFP+ DRGs at 72 hpf after genetic inhibition of Wnt/β-catenin signaling. *ngn1*:GFP+ DRGs were counted on one side of the larvae. (**F**) Maximal intensity projection of a confocal z-stack of the cranial vasculature and quantification of hindbrain CtAs in control or *gpr124* morphant *Tg(kdrl:EGFP)* embryos after exposure to the indicated GSK-3β inhibitors from the 16-somite stage

*Figure 3. continued on next page*

*Figure 3. Continued*

onwards. Red arrows point to CtAs. Pharmacological inhibitions were performed as described in 'Methods'.
(**G**) Dorsal views and quantification of DRGs in control or *gpr124* morphant *Tg(ngn1:GFP);Tg(sox10:mRPF)* embryos after exposure to the indicated GSK-3β inhibitors from the 16-somite stage onwards. *ngn1*:GFP[+] DRGs were counted on one side of the larvae. Pharmacological inhibitions were performed as described in 'Methods'. Scale bars, 50 µm. In all panels, values represent means ± SD (*p < 0.05; **p < 0.01; Kruskal–Wallis test: (**D**), (**F**), and (**G**); Mann–Whitney test: (**E**)).

To test whether the combination of Reck and Gpr124 can co-activate signaling by Wnts other than Wnt7a and Wnt7b, we screened all 19 Wnts and Norrin for co-activation by Gpr124 alone, Reck alone, or Gpr124 plus Reck (*Figure 4D*, *Figure 4—figure supplement 1B, C*). This experiment shows that the highest co-activation with Gpr124 plus Reck occurs with Wnt7a and Wnt7b, with other Wnts and Norrin showing little or no response. An analogous comparison among the ten Frizzled receptors showed that Gpr124/Reck stimulates Wnt7a- and Wnt7b-mediated signaling by multiple Frizzleds (*Figure 4C*, *Figure 4—figure supplement 1A*). The data predict that cells expressing Gpr124, Reck, Lrp5 and/or Lrp6, and any of multiple Frizzleds will be responsive to Wnt7a or Wnt7b.

In earlier experiments, we observed that Gpr124 co-activation of Wnt7a- and Wnt7b-dependent signaling in STF cells could be stimulated by Lrp5 but not by Lrp6, and we postulated that one or more additional proteins might be required to enhance signaling in the presence of Lrp6 (*Zhou and Nathans, 2014*). The present experiments identify Reck as the missing protein since signaling in the presence of Reck can be stimulated by both Lrp5 and Lrp6 (*Figure 4A*), and they provide functional evidence that Gpr124 and Reck are major determinants of both the amplitude and ligand specificity of Wnt/β-catenin signaling.

## Reck and Gpr124 expressed in 293 cells interact at the plasma membrane

To examine whether the functional interactions between Gpr124, Reck and the Fz/Lrp5/6 complex reflect a multi-component membrane receptor complex involved in Wnt7 binding, their subcellular localization was investigated in 293 cells. We generated amino-terminal epitope-tagged version of Gpr124 (FLAG-Gpr124) and Reck (HA-Reck) and validated the biological activity of the fusion proteins in STF assays (data not shown) and brain vascular rescue experiments in zebrafish embryos (*Figure 5A*). When expressed in 293 cells, FLAG-Gpr124 and HA-Reck colocalized at the plasma membrane as revealed by indirect immunofluorescence assays (*Figure 5B*). Both Gpr124 and Reck reached and resided at the plasma membrane irrespective of the presence of the other (*Figure 5B*), and within this compartment they co-localized with GFP-tagged Fz4 (*Figure 5—figure supplement 1*). We next evaluated the proximity of Gpr124 and Reck through a highly sensitive in situ proximity ligation assay (PLA) which allows the localized detection of protein interactions (*Söderberg et al., 2006*). While PLA assays on 293 cells expressing either FLAG-Gpr124 or HA-Reck individually remained negative, simultaneous expression of the fusion proteins yielded extensive fluorescence hybridization signals at the cell surface (*Figure 5C, D*). As an additional control, the FLAG-Dvl2/HA-Reck protein pair, generated weaker and less frequent PLA signals, despite increased anti-FLAG immunoreactivity at the plasma membrane (*Figure 5C*). These observations, coupled with the synergistic capacity of Gpr124 and Reck to stimulate and confer ligand specificity to Wnt/β-catenin signaling in a Frizzled and Lrp5/6-dependent manner, are compatible with the assembly of a multi-component Wnt7 membrane receptor complex, whose precise stoichiometry and composition remains to be determined (*Figure 5E*, see also 'Discussion').

## Wnt/β-catenin signaling cell-autonomously controls tip cell function

In contrast to mouse, the size of zebrafish embryos allows oxygen to reach tissues independently of cardiovascular-based convection. We performed microarray gene expression analysis and found, in line with normoxic conditions, that the transcript levels of various *vegf* genes were not increased in the absence of Gpr124 at 48 hpf (*Figure 6A*). We took advantage of the unique normoxic nature of zebrafish embryos, combined with their optical clarity to investigate the requirements for Wnt/β-catenin during CNS angiogenesis at single-cell resolution.

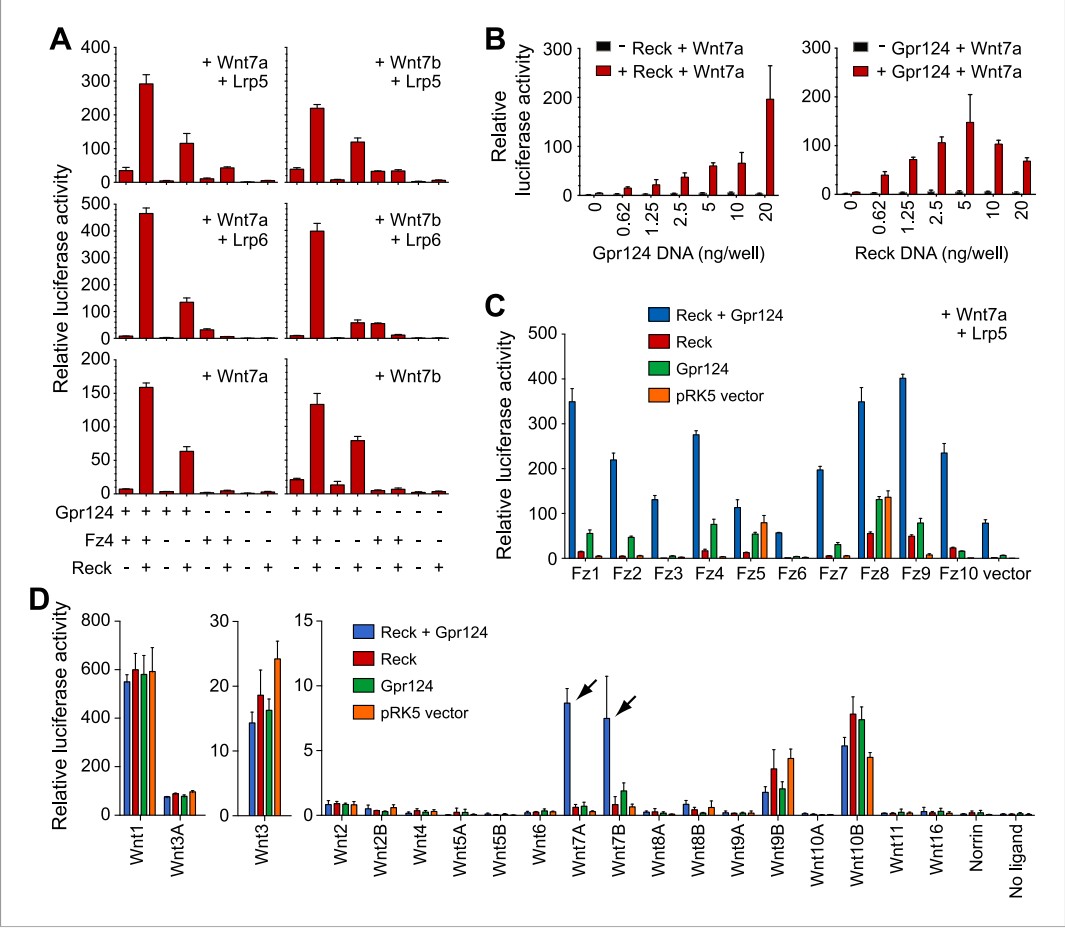

**Figure 4**. Synergy between Gpr124 and Reck in co-activating Wnt7a- and Wnt7b-dependent Wnt/β-catenin signaling in cell culture. (**A**) Plasmids encoding Wnt7a (left) or Wnt7b (right), together with Lrp5 (top row), Lrp6 (center row), or no Lrp (bottom row) were transfected with Gpr124, Fz4, and/or Reck plasmids as indicated. (**B**) Titration of Gpr124 with or without Reck (left), and titration of Reck with or without Gpr124 (right). (**C**) Comparison of the ten mouse Frizzleds transfected with Lrp5 and Wnt7a, with or without Gpr124 and/or Reck. (**D**) Comparison of the nineteen mouse Wnts and Norrin, transfected with or without Gpr124 and/or Reck. Arrows point to Wnt7a and Wnt7b responses. Luciferase assays in transiently transfected STF cells were performed as described in 'Methods'. Values represent means ± SD.

The following figure supplement is available for figure 4:

**Figure supplement 1**. Synergy between Gpr124 and Reck in co-activating Wnt7a- and Wnt7b-dependent Wnt/β-catenin signaling in cell culture.

We first asked whether defective brain vascularization might result from insufficient EC abundance in the parental vessels. Using the endothelial-specific nuclear reporter *Tg(kdrl:NLS-GFP)* (*Blum et al., 2008*), we determined EC number in the PHBC at 30 hpf and observed no difference between *gpr124* mutants and their wild-type siblings (*Figure 6B*). As expected, marked differences were detected after the onset of brain invasion; while wild-type siblings accumulated intra-cerebral ECs contributing to the CtAs, *gpr124* mutants maintained marginally higher numbers of cells in the PHBC, BA and most prominently in ventral connections between the PHBCs and the BA (*Figure 6—figure supplement 1*, *Figure 1D*, *Videos 1 and 2*). Mammalian CNS vessels show extensive pericyte coverage and these support cells convey essential properties to the BBB (*Armulik et al., 2010*; *Bell et al., 2010*; *Daneman et al., 2010*). To determine whether EC-pericyte interactions were compromised in *gpr124* mutants, we generated a Pdgfrb transgenic reporter line *TgBAC(pdgfrb:mCitrine)* that labels perivascular cells presenting all characteristics of *bona fide* pericytes. We found extensive vascular wall coverage in

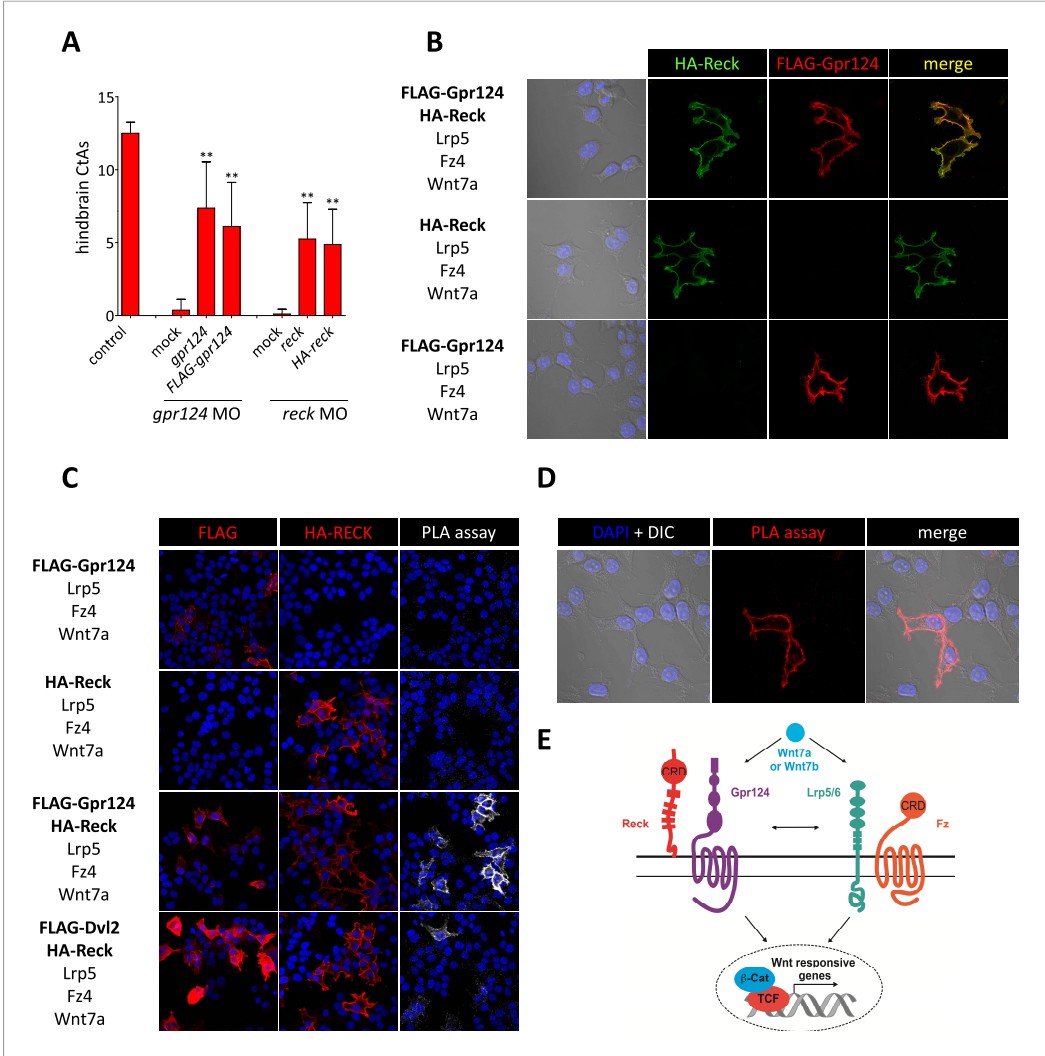

**Figure 5**. Interaction between Gpr124 and Reck in cultured cells. (**A**) Quantification of hindbrain CtAs in control and *gpr124* morphants at 60 hpf after injection at the one-cell stage of 100 pg RNA encoding Gpr124, Reck or their epitope-tagged versions, FLAG-Gpr124 and HA-Reck. (**B**) and (**C**) Plasmids encoding FLAG-Gpr124, HA-Reck, Lrp5, Fz4, and Wnt7a were transfected in 293 cells as indicated. 48 hours after transfection, epitope-tagged fusion proteins were detected by indirect immunofluorescence (**B**) or by proximity ligation assays as described in 'Methods'. (**D**) High magnification views of PLA signals between FLAG-Gpr124 and HA-reck. (**E**) Illustration of the plasma membrane protein complexes that mediate Wnt7a or Wnt7b signaling during CNS angiogenesis. Values represent means ± SD (*p < 0.05; **p < 0.01; Kruskal–Wallis test).

The following figure supplement is available for figure 5:

**Figure supplement 1**. Gpr124, Reck, and Fz4 colocalize at the plasma membrane in cultured cells.

both the presence and absence of Gpr124 function (*Figure 6C*). We then examined EC behaviors using time-lapse confocal microscopy in wild-type and Wnt/β-catenin-deficient vessels during CNS angiogenesis. Dynamic filopodial extensions were observed from the PHBC of both wild-type and *gpr124* mutants embryos (*Figure 6D*, *Videos 3–5*), but only in wild-type siblings did those extensions progress dorsally to allow the cell body of a stereotypic tip cell (*Gerhardt et al., 2003*) to emerge from extra-cerebral vessels and invade the brain tissue. Intra-cerebral EC nuclei were absent in *gpr124* mutants (*Videos 6 and 7*). Similar results were seen upon Reck knockdown as well as Wnt/β-catenin inhibition by heat-shock induced Axin1 overexpression.

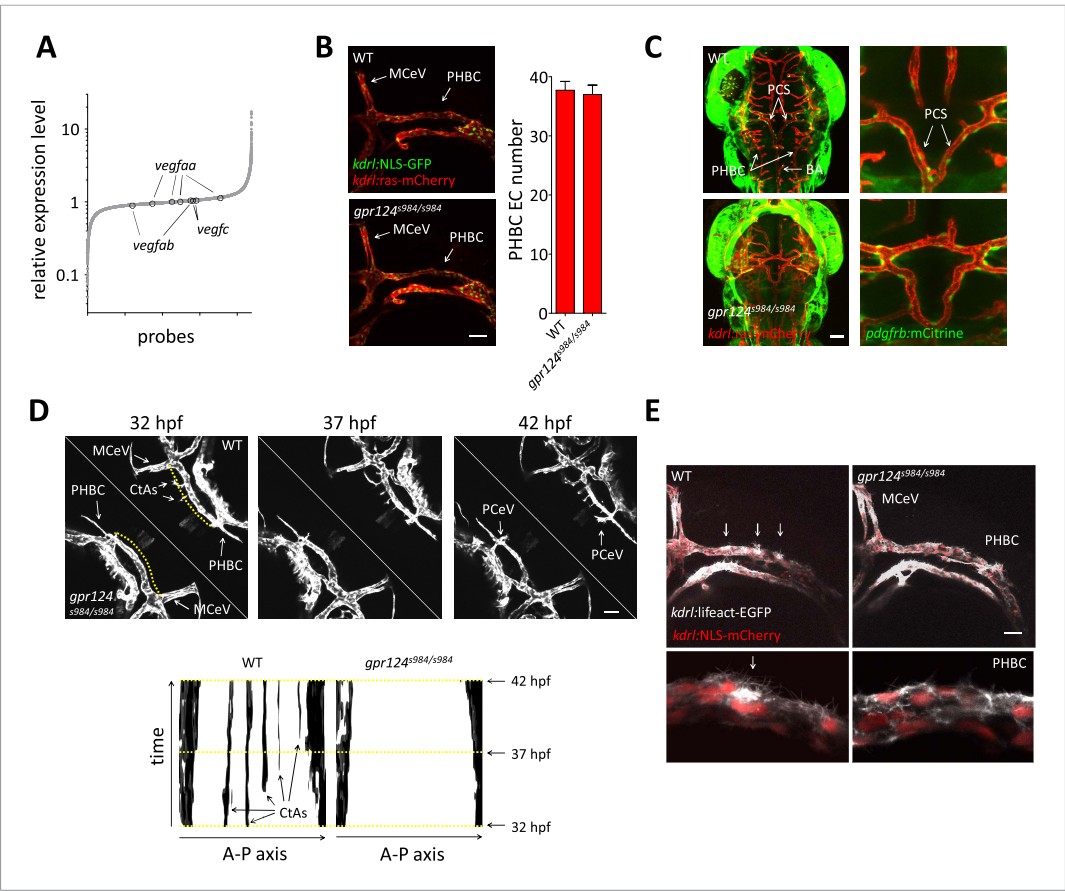

**Figure 6**. Tip cell defects in Wnt/β-catenin-deficient PHBCs. (**A**) Scatterplot of a microarray comparison of wild-type and *gpr124^{s984/s984}* embryos at 48 hpf. Each dot refers to the relative signal intensity of a given probe in mutant vs wild-type embryos. Circles identify independent probes for *vegfaa*, *vegfab*, and *vegfc*. (**B**) Endothelial cell number in the PHBCs of wild-type and *gpr124^{s984/s984}* embryos at 30 hpf. *kdrl*:NLS-GFP^+ nuclei were counted on one side of the embryos. (**C**) Dorsal views of *Tg(pdgfrb:mCitrine);Tg(kdrl:ras-mCherry)* wild-type and *gpr124^{s984/s984}* embryos at 40 hpf. (**D**) Stills from *Video 3* recording sprouting angiogenesis in the hindbrain of *Tg(kdrl:GFP)* wild-type and *gpr124^{s984/s984}* embryos. Below are kymograph plots showing a time course of pixel intensity across the yellow virtual lines running 15 μm dorsal to the PHBC (from the anterior MCeV to the posterior PCeV [posterior cerebral vein]) and depicted in the upper left panel. (**E**) Lateral views of *Tg(kdrl:lifeact-GFP);Tg(kdrl:NLS-mCherry)* wild-type and *gpr124^{s984/s984}* hindbrains at 30 hpf. Arrows point to actin-dense structures forming in the PHBC vessel wall. Scale bars, 50 μm. Values represent means ± SD.

The following figure supplement is available for figure 6:

**Figure supplement 1**. Endothelial cell number in wild-type and *gpr124^{s984/s984}* brain vessels at 54 hpf.

Polar filopodial extensions and abluminal emergence of ECs from pre-existing vessels are easily detectable angiogenic events occurring after the VEGF/Notch-controlled (*Hellström et al., 2007*; *Leslie et al., 2007*; *Lobov et al., 2007*; *Siekmann and Lawson, 2007*; *Suchting et al., 2007*; *Phng and Gerhardt, 2009*; *Eilken and Adams, 2010*) specification of presumptive tip and stalk cells within the parental vessel. We sought to distinguish between a defect in filopodial extension and an earlier event occurring within the parental vessel, by contrasting F-actin structures within the pre-angiogenic PHBCs of wild-type and *gpr124* mutant embryos using a novel *Tg(kdrl:lifeact-EGFP)* line that labels filamentous actin (*Riedl et al., 2008*; *Phng et al., 2013*) in ECs. While the use of this transgene confirmed the presence of filopodial extensions independently of Gpr124 function, wild-type siblings in addition displayed actin-rich structures at the dorsal wall of the PHBCs at sites where angiogenic tip cells later emerged. These structures were not observed in *gpr124* mutants, indicating that Gpr124

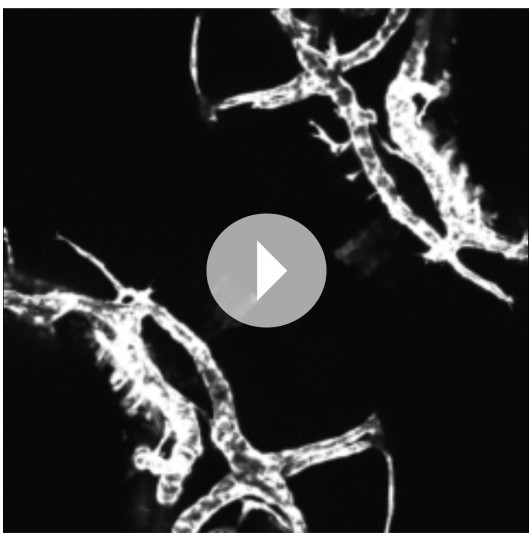

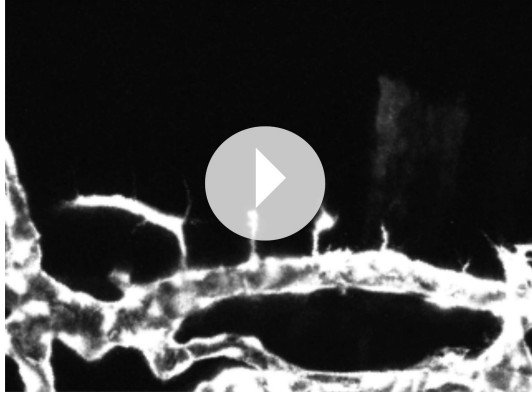

**Video 4.** Hindbrain angiogenesis in wild-type embryos. High magnification view of the time-lapse confocal video presented in *Video 3*, focusing on the wild-type hindbrain (anterior to the left).

**Video 3.** Brain angiogenesis in wild-type and *gpr124$^{s984/s984}$* embryos. Time-lapse confocal video of *Tg(kdrl:GFP)* wild-type (upper right) and *gpr124$^{s984/s984}$* (bottom left) embryos, starting at 32 hpf and ending at approximately 42 hpf.

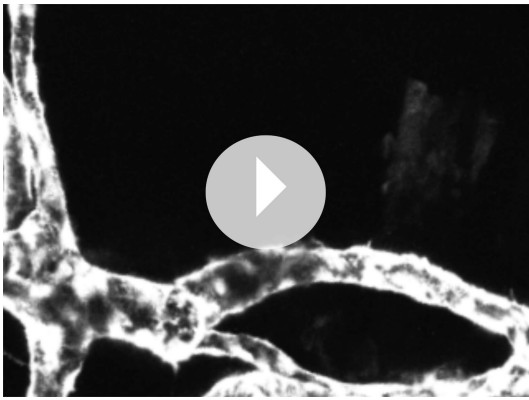

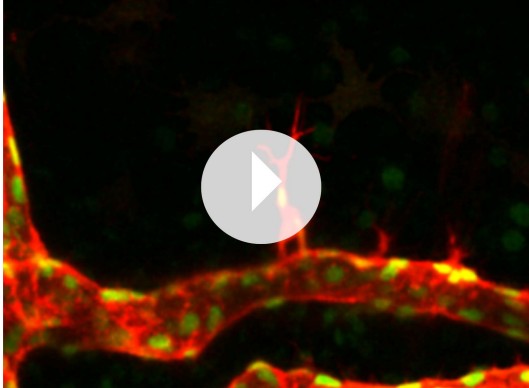

**Video 6.** Endothelial cell invasion into the wild-type hindbrain. Time-lapse confocal video generated from maximum intensity confocal projections through one hindbrain hemisphere of a *Tg(kdrl:NLS-GFP);Tg(kdrl:ras-mCherry)* wild-type embryo. Endothelial cell nuclei are green (GFP); endothelial cells are red (mCherry). Video starts at 32 hpf and ends at approximately 42 hpf (anterior to the left).

**Video 5.** Hindbrain angiogenesis in *gpr124$^{s984/s984}$* embryos. High magnification view of the time-lapse confocal video presented in *Video 3*, focusing on the *gpr124$^{s984/s984}$* hindbrain (anterior to the left).

regulates actin cytoskeletal rearrangement within extra-cerebral ECs before the onset of brain invasion (*Figure 6E*).

These observations are consistent with a defect in tip cell specification or behavior within the PHBCs of *gpr124* mutants. To test this hypothesis, we generated genetically mosaic PHBCs by cell transplantation at mid-blastula stages and examined the behavior of wild-type *Tg(kdrl:GFP)* ECs in the context of *Tg(kdrl:ras-mCherry) gpr124*, *reck* or Wnt/β-catenin deficient endothelial neighbors (*Figure 7A, B*). Time-lapse imaging (*Video 8* and stills presented in *Figure 7A*) revealed that mosaic PHBCs were competent for brain vascular invasion and that in all sprouts examined, tip cells were wild-type (green). Notably, trailing stalk cells could be wild-type (green) or Gpr124/Reck/Wnt/β-catenin-deficient (red) (red arrows in *Figure 7A*) revealing a tip cell-autonomous role for Gpr124/Reck/Wnt/β-catenin signaling during brain angiogenesis. A single wild-type cell was sufficient to

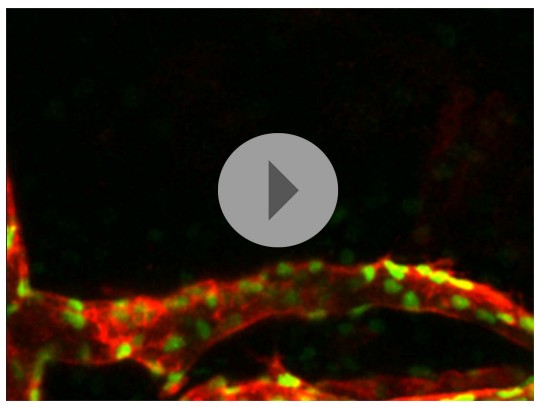

**Video 7.** Lack of endothelial cell invasion into the *gpr124* mutant hindbrain. Time-lapse confocal video generated from maximum intensity confocal projections through one hindbrain hemisphere of a *Tg(kdrl:NLS-GFP);Tg(kdrl:ras-mCherry) gpr124s984/s984* embryo. Endothelial cell nuclei are green (GFP); Endothelial cells are red (mCherry). Video starts at 32 hpf and ends at approximately 42 hpf (anterior to the left).

instruct Wnt/β-catenin-deficient ECs to assemble mosaic vessels (*Video 9*). These mosaic sprouts invariably led by wild-type tip cells (*Figure 7C*) invaded the hindbrain in a wild-type manner and lumenized after connecting medially to the BA (*Figure 7B, D*). In contrast, when wild-type *Tg (kdrl:GFP)* cells were transplanted into *Tg(kdrl: ras-mCherry)* wild-type hosts, tip cell genotypes were randomized (*Figure 7C*). In mosaic ISV sprouts, *gpr124* or *reck* loss of function did not impact tip cell genotype (*Figure 7E, F*). The mosaic cerebral vessels were maintained for several days with no indication of vascular in-stability during embryonic or early larval development (*Figure 7G*).

The tip cell-restricted requirement for Gpr124 and Reck mediated Wnt/β-catenin signaling could conceptually be linked to a role in initiating a tip cell-permissive arterial-biased transcrip-tional program (*Corada et al., 2010*) that is known to operate in the PHBCs during brain angiogenesis (*Bussmann et al., 2011*) and that results notably in *dll4* expression. At 36 hpf however, *dll4* transcripts could be detected in the PHBCs irrespective of Gpr124 function (*Figure 7—figure supplement 1*) and accordingly, mosaic endothelial-specific overexpression of Dll4 was not sufficient to restore brain angiogenic competence to Gpr124-deficient PHBCs, while in a similar setting Gpr124-positive tip cells led new brain vascular sprouts (*Figure 7—figure supplement 1*).

## Discussion

The principal results of the present study are: (1) characterization of the pattern of Wnt/β-catenin signaling in the developing zebrafish brain vasculature and the consequences of reduced Wnt/β-catenin signaling on brain angiogenesis, (2) identification of Reck, together with Gpr124, as integral components of a novel Wnt7a/Wnt7b-specific receptor complex in both zebrafish and mammals, and (3) discovery of a distinctive requirement for Wnt/β-catenin signaling in tip cells during brain angiogenesis.

### Wnt/β-catenin signaling in tip cells

During retinal vascular development in mouse, wild-type ECs can instruct *Fz4*⁻/⁻ ECs to assemble into mosaic vessels, suggesting that Wnt/β-catenin signaling (*Wang et al., 2012*) is not a uniform requirement for all ECs during vascular network formation. The cellular basis of this phenomenon has, until now, not been investigated. Using genetic mosaics and live imaging, we present evidence that the control of brain vascular invasion by Wnt/β-catenin signaling operates at the level of the tip cells, and that stalk cells that are deficient in Wnt/β-catenin signaling can follow wild-type tip cells and contribute to the developing vasculature (*Figure 7H*). These observations reveal a heterogeneous requirement for Wnt/β-catenin signaling among ECs during brain invasion in zebrafish, and they predict a similar differential requirement between tip and stalk cells in the mammalian brain and retina.

In recent years, through the investigation of a limited number of stereotypical in vivo settings, most notably the postnatal mouse retina and the zebrafish ISVs, a coherent model of sprouting angiogenesis integrating controlled behaviors of VEGF-selected tip cells and Notch-induced stalk cells within nascent vessels has emerged. A question of pressing interest is whether tissue and organ-specific angiogenic programs might refine our understanding of these processes. The Wnt/β-catenin-dependency of brain angiogenesis in zebrafish (*Figure 3*) coupled to their unique optical attributes has permitted us, through live imaging of mosaic animals (*Figure 7*), to collect evidence that local angiogenic programs can impact directly on the basic cellular behaviors of nascent angiogenic sprouts.

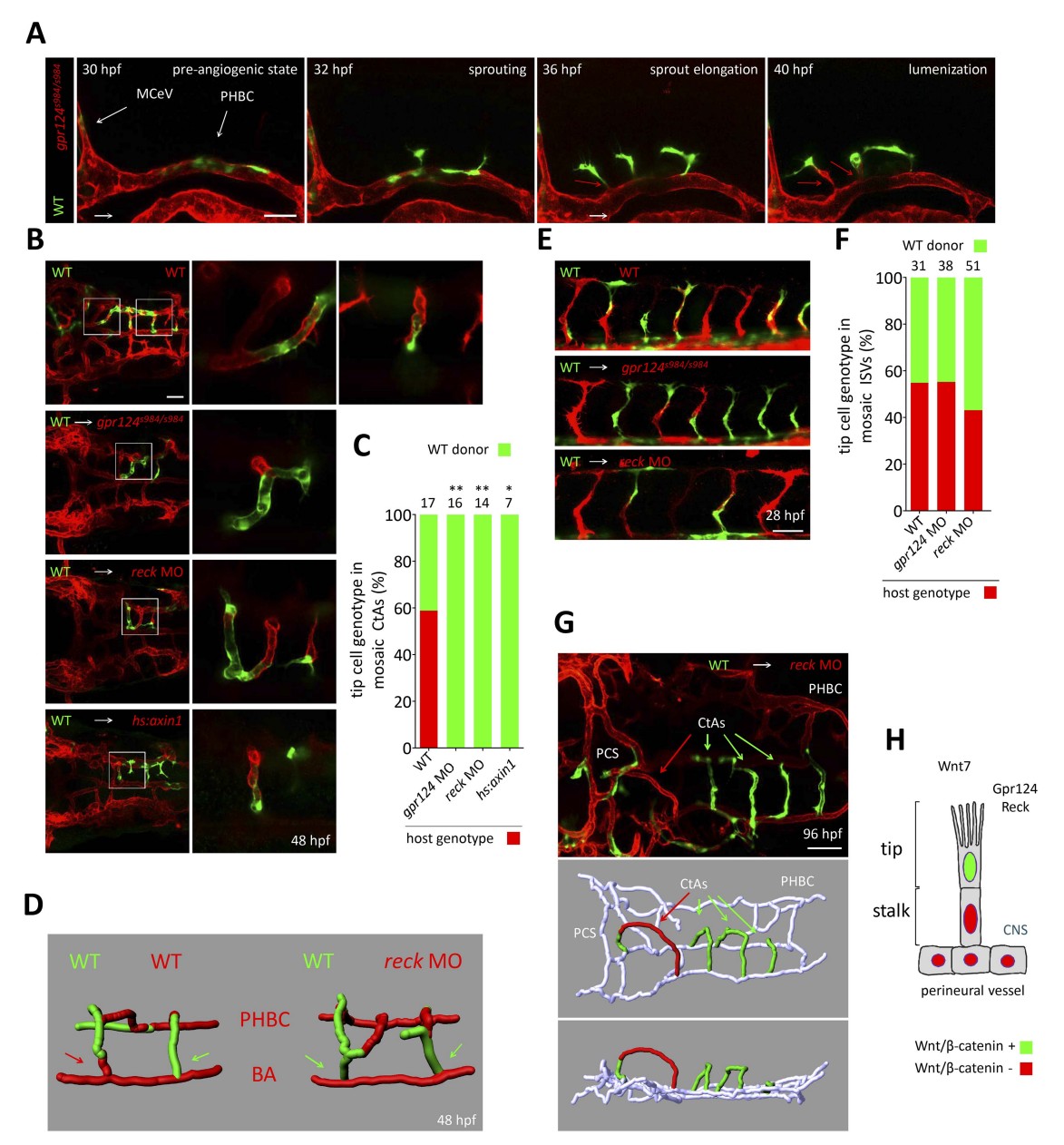

**Figure 7**. Tip cell-specific requirement for Gpr124 and Reck-controlled Wnt/β-catenin signaling. (**A**) Stills from *Video 8* recording sprouting angiogenesis from a mosaic PHBC obtained by blastula-stage transplantation. Green *kdrl*:GFP⁺ endothelial cells derive from a wild-type donor embryo, red *kdrl*:ras-mCherry⁺ endothelial cells are from the *gpr124^s984/s984* host. (**B**) Dorsal views of 48 hpf mosaic cranial vasculatures of the indicated genotypes. Right panels are high magnification views of a confocal z-stack of relevant depth illustrating intra-cerebral mosaic CtAs corresponding to the boxed areas in the left panels. (**C**) Contribution of cells of defined genotype to the tip cell position of mosaic CtAs after transplantation of wild-type *Tg(kdrl:GFP)* donor cells (green) into *Tg(kdrl:ras-mCherry)* host blastulae of the indicated genotype (red). Number of mosaic vessels analyzed is indicated above each bar. (**D**) 3D Imaris (Bitplane) reconstruction of representative hindbrain mosaic vascular networks. Arrows point to the cells connecting with the BA, after leading the CtA as tip cells. (**E**) Lateral views of the trunk region after transplantation of *Tg(kdrl:GFP)* wild-type donor cells into *Tg(kdrl:ras-mCherry)* host blastulae of the indicated genotype. (**F**) Contribution of cells of defined genotype to the tip cell position of mosaic ISVs after transplantation of wild-type *Tg(kdrl:GFP)* donor cells (green) into *Tg(kdrl:ras-mCherry)* host blastulae of the indicated genotype (red). Number of mosaic vessels analyzed is indicated above each bar. (**G**) Dorsal view of the cranial vasculature and 3D Imaris (Bitplane) reconstruction of the brain vessels of a 96 hpf mosaic larva after transplantation of wild-type *Tg(kdrl:GFP)* donor cells into *reck* morphant *Tg(kdrl:ras-mCherry)* host blastula. (**H**) Cellular requirement for Wnt/β-catenin, Gpr124, and Reck during sprouting angiogenesis in the zebrafish CNS (*p < 0.05; **p < 0.01; exact Fisher test).

The following figure supplement is available for figure 7:

*Figure 7. continued on next page*

*Figure 7. Continued*

**Figure supplement 1**. Transgenic endothelial *dll4* expression is not sufficient to rescue the *gpr124* mutant vascular defects.

An important focus for future studies will be to delineate the molecular mechanisms by which Wnt/β-catenin signaling affects tip cell specification and/or behavior. According to the current model, when pro-angiogenic signals stimulate a quiescent vessel, ECs that experience the highest level of VEGFA-VEGFR2 signaling become tip cells. As these nascent tip cells egress from the parental vessel, they accumulate tip cell-enriched transcripts, including *dll4* and *vegfr3*, while neighboring cells become stalk cells through Notch-mediated lateral inhibition (*Hellström et al., 2007*; *Leslie et al., 2007*; *Lobov et al., 2007*; *Siekmann and Lawson, 2007*; *Suchting et al., 2007*). The multiple interactions between the Wnt/β-catenin, Notch, and VEGFR signaling systems (*Phng et al., 2009*; *Corada et al., 2010*; *Gore et al., 2011*), as seen, for example, by the strong reduction of hindbrain CtAs after loss of VEGFR2 signaling or Notch activation (*Bussmann et al., 2011*), imply that regulation of brain vascularization is highly integrated.

Despite current evidence for the tight integration of multiple signaling systems in EC development, our data argue that the tip cell requirement for Wnt/β-catenin signaling operates through a mechanism that does not simply reflect a global modulation of VEGF and/or Notch signals. In *gpr124* mutants and *reck* morphants, a causal role for reduced VEGFR2 signaling in the PHBC appears unlikely because the formation of the BA from the PHBCs, which is known to be sensitive to VEGF inhibition (*Bussmann et al., 2011*), appears unaffected. We have also shown that at 36 hpf *dll4* transcripts could be detected in *gpr124* mutant PHBCs as in wild-type, and that clonal endothelial *dll4* overexpression was not sufficient to restore the angiogenic behavior. With respect to Notch signaling, although *gpr124* and *reck*-depleted cerebral vasculatures exhibit defects resembling those generated by pan-endothelial expression of a constitutively active Notch intracellular domain (NICD) (*Bussmann et al., 2011*), *gpr124* and *reck* morphant vascular defects were not rescued by *dll4* knock-down or gamma-secretase inhibitor (DAPT) treatment.

In our experiments, the Wnt reporter transgene labels both tip and stalk cells in nascent sprouts, as well as ECs in adult CNS vessels (*Moro et al., 2012*), reflecting the role of Wnt/β-catenin signaling in

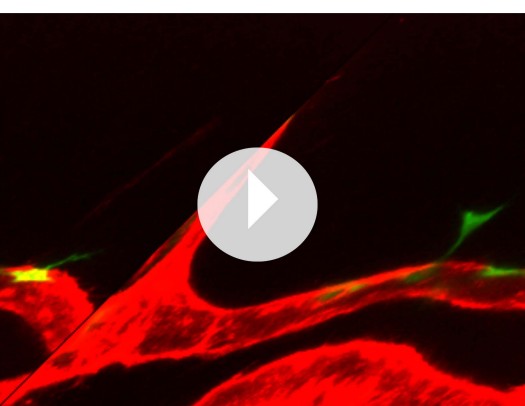

**Video 8.** Single-cell analysis of CtA formation in *gpr124* mosaic animals. CtAs are led by wild-type tip cells and *gpr124* mutant cells can incorporate into the sprouts as stalk cells. Time-lapse confocal video generated from maximum intensity confocal projections through one hindbrain hemisphere of a mosaic embryo containing wild-type *kdrl*:GFP⁺ ECs (green) and *gpr124*ˢ⁹⁸⁴/ˢ⁹⁸⁴ *kdrl*:ras-mCherry⁺ ECs (red). Video starts at 30 hpf and ends at approximately 40 hpf (anterior to the left).

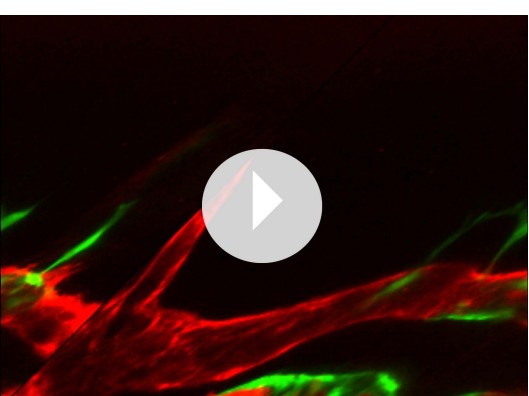

**Video 9.** Single-cell analysis of CtA formation in *gpr124* mosaic animals. A single wild-type tip cell is sufficient to initiate the formation of an intra-cerebral CtA. Time-lapse confocal video generated from maximum intensity confocal projections through one hindbrain hemisphere of a mosaic embryo containing wild-type *kdrl*:GFP⁺ ECs (green) and *gpr124*ˢ⁹⁸⁴/ˢ⁹⁸⁴ *kdrl*:ras-mCherry⁺ ECs (red). Video starts at 30 hpf and ends at approximately 40 hpf (anterior to the left).

angiogenesis, BBB acquisition, and BBB maintenance (*Reis et al., 2012*; *Wang et al., 2012*). The multiple roles of Wnt/β-catenin signaling in endothelial development complicate the interpretation of experiments designed to discriminate between a permissive role for Wnt/β-catenin signaling during CNS angiogenesis vs a selective role in tip cell differentiation. For example, Wnt/β-catenin signaling could regulate a transcriptional program that is required for tip cell function but plays no role in tip cell selection or, alternatively, tip cells could be selected through a specific transcriptional program that is only initiated by a certain threshold level of Wnt/β-catenin activity. Although the present data do not distinguish between these two scenarios, we note that Wnt/β-catenin reporter expression does not begin uniformly within PHBCs at 26–28 hpf, but instead labels a small number of cells, and even at later stages (32 hpf), different levels of Wnt/β-catenin reporter expression are observed among PHBC ECs (*Figure 3A*). It is also notable that sprouting tip cells display high levels of Wnt/β-catenin reporter expression.

## Gpr124 and Reck: ligand specificity factors in Wnt/β-catenin signaling

Our discovery of Reck's central role in Wnt/β-catenin signaling is surprising because earlier work had implicated Reck in tissue remodeling via inhibition of MMP function at the transcriptional level (*Takagi et al., 2009*), and at the level of protein–protein interactions (*Takahashi et al., 1998*). Down-regulation of RECK has been shown to correlate with enhanced tumor invasion, angiogenesis, and metastasis (*Noda and Takahashi, 2007*), and *Reck*$^{-/-}$ mouse embryos die in utero at around E10.5 with prominent vascular defects including vessel dilation, hemorrhaging, and arrested development of the primary vascular plexus (*Oh et al., 2001*; *Chandana et al., 2010*). The mechanisms underlying these vascular defects have been unclear, but in light of the present work, a contribution of defective Wnt/β-catenin signaling seems possible.

A question of central importance is the mechanism by which Gpr124 and Reck promote ligand-specific Wnt/β-catenin signal transduction. It is possible that Reck's role as an MMP inhibitor may contribute to its activity in the context of Wnt/β-catenin signaling. For example, Reck might protect ECM- or membrane-bound Wnt ligands and/or receptor complexes from proteolytic degradation, a possibility that would be consistent with its capacity to modulate Notch ligand ectodomain shedding through ADAM10 antagonism during cortical neurogenesis (*Muraguchi et al., 2007*). In the context of this model, Gpr124 itself would be an attractive candidate for Reck-dependent regulation given previous evidence that MMP-processing of Gpr124 might affect its function (*Vallon and Essler, 2006*). Arguing against this model is our failure to rescue *reck* morphant vascular or DRG phenotypes by treating embryos with various MMP or ADAM inhibitors (data not shown), in agreement with results from a previous study that examined Reck's role in DRG development (*Prendergast et al., 2012*). We note that Reck has additionally been implicated in endocytic trafficking (*Miki et al., 2007*), focal adhesion formation and cell polarity (*Morioka et al., 2009*) processes that may affect Wnt/β-catenin signal transduction (*Fonar and Frank, 2011*; *Feng and Gao, 2014*).

The striking similarities between Gpr124 and Reck loss-of-function phenotypes, together with their synergistic roles in Wnt7a/Wnt7b-specific activation of Wnt/β-catenin signaling, suggest a more focused view of Reck function. As an extension of a previous model for Gpr124 action (*Zhou and Nathans, 2014*), we propose that Reck and Gpr124 cooperate to activate Wnt/β-catenin signaling by selectively promoting Wnt7a/Wnt7b binding to cell surface receptor complexes, thereby (1) conferring specificity among Wnt ligands that would not otherwise be discriminated by Frizzled-binding (*Janda et al., 2012*), and (2) greatly enhancing the amplitude of Wnt7a and Wnt7b signals. Moreover, the observation that maximal Wnt/β-catenin signaling activity in Wnt7a/Wnt7b-stimulated cells requires the simultaneous presence of Gpr124, Reck, Fz, and Lrp5/Lrp6 (*Figure 4*), suggests that the Wnt7a/Wnt7b receptor complex might be composed of all four proteins (*Figure 5E*). An alternative model is suggested by the intriguing observation that Reck contains a cysteine-rich domain that is homologous to the Wnt-binding cysteine-rich domain of Frizzled receptors (*Pei and Grishin, 2012*) and by our demonstration that the intracellular domain of Gpr124 is functionally interchangeable with the homologous domain of Gpr125 which has been shown to bind the cytoplasmic Wnt signaling adaptor Dishevelled (*Li et al., 2013*; *Figure 1F*). Taken together, these two observations suggest that Gpr124 and Reck might be capable of signaling as an autonomous Wnt-binding receptor complex (*Figure 5E*).

## Perspective

Vascular morphogenesis is a complex process that requires coordinated control of EC behaviors, including a separation of ECs into leading tip cells and trailing stalk cells. In addition, ECs must respond to organ-specific signals to meet the specialized needs of the vascularized tissue. By observing CNS angiogenesis in genetic mosaics at single-cell resolution, we have identified a regulatory system whereby endothelial Wnt/β-catenin signaling controls CNS angiogenesis through selective modulation of tip cell function. This Wnt/β-catenin signaling pathway requires the membrane proteins Gpr124 and Reck, further expanding the complexity of Wnt signal transduction mechanisms and holding the promise of new insights into the etiology and treatment of cerebral vascular defects, a major cause of morbidity and mortality.

## Materials and methods

### Zebrafish husbandry and transgenic lines used

Zebrafish were maintained and embryos were obtained and raised under standard conditions. Establishment and characterization of the Tg(kdrl:GFP)$^{s843}$, Tg(kdrl:ras-mCherry)$^{s896}$, Tg(kdrl:NLS-GFP)$^{zf109}$, Tg(hsp70l:dkk1-GFP)$^{w32}$, Tg(hsp70l:Mmu.Axin1-YFP)$^{w35}$, Tg(7xTCF-Xla.Siam:GFP)$^{ia4}$, Tg(ngn1:gfp), Tg(UAS: EGFP), Tg(sox10:mRFP)$^{vu234}$, Tg(kdrl:NLS-mCherry)$^{is4}$ transgenic lines have been described elsewhere (*Blader et al., 2003*; *Jin et al., 2005*; *Kirby et al., 2006*; *Stoick-Cooper et al., 2007*; *Asakawa and Kawakami, 2008*; *Blum et al., 2008*; *Chi et al., 2008*; *Wang et al., 2010*; *Kagermeier-Schenk et al., 2011*; *Moro et al., 2012*). The Tg(fli1:Gal4db-TΔC-2A-mC) (*Kashiwada et al., 2015*) was generated by fusing a cDNA fragment encoding the β-catenin binding domain of human TCF4E (amino acid 1–314) to the DNA binding domain of Gal4 followed by a 2A peptide and mCherry. It was subcloned into the pTol2-fli1 vector to generate the pTol2-fli1:Gal4db-TΔC-2A-mC plasmid. The Tg(fli1:Myr-mCherry) was generated by fusing an oligonucleotide encoding the myristoylation (Myr) signal derived from Lyn to mCherry in the pTol2-fli1 vector, yielding the pTol2-fli1:Myr-mC plasmid. The Tg (kdrl:Lifeact-EGFP)$^{s982}$ line was generated following cloning of the F-actin binding Lifeact-EGFP downstream of a 6.8 kb kdrl promoter in pTol2. The TgBAC(pdgfrb:citrine)$^{s1010}$ line was generated using a BAC clone containing the start codon of pdgfrb (CH73-289D6) using a pRedET based protocol (*Bussmann and Schulte-Merker, 2011*). Tol2 LTRs were inserted into the vector backbone. Recombination was performed by amplifying a mCitrine-Kan(R) cassette using primers that contained 50bp of homology flanking the start codon: pdgfrb_HA1_mCitrine, TTTGGCTTTGAGGCGAATCAGT CATGTTGTTTTCTCTCCGTCTGCAGTGTACCATGGTGAGCAAGGGCGAGGAG and pdgfrb_HA2_Kan(R), TGGATGCGGCTGATGGTCGAACTCTTCATGCTTTCTTCTAGAGCAGGACATCAGAAGAACTCGTC AAGAAGGCG. Zebrafish transgenic lines were generated according to previously described protocols (*Kawakami, 2004*).

### Imaging

All images were acquired using a Leica (Wetzlar, Germany) M165 stereomicroscope or a Zeiss (Oberkochen, Germany) LSM710 confocal microscope after embryo anesthesia with a low dose of tricaine and immobilization in 1% low-melting agarose in glass-bottom Petri dishes (MatTek Corporation, Ashland, MA). Time-lapse images were recorded every 10 min for a total period of 16 hr. The microscope stage was enclosed in a temperature-controlled chamber, and samples were kept at 28.5°C. The Sulfo-NHS-Biotin leakage assay was performed by intra-ventricular injection of 5 nL of 0.5 mg/mL EZ-link Sulfo-NHS-Biotin (Pierce; 0.5 kDa) in D-PBS, with the help of a micromanipulator, in animals treated with a low dose of tricaine. After 60 min, animals were anaesthetized with an overdose of tricaine, and the brains and livers were dissected and fixed in 4% PFA overnight at 4°C before embedding in 4% agarose. Vibratome sections of 150 μm thickness were permeabilized in 0.4% PBT (Triton X-100) and blocked in 4% PBTB (BSA). Primary and secondary antibody staining was performed at 4°C overnight. The following primary antibodies were used: anti-GLUT1 (NB300-666; Novus Biologicals, Littleton, CO), anti-P Glycoprotein [C219] (ab3364; Abcam, Cambridge, UK). Biotin was detected with Alexa Fluor 647 streptavidin. In all confocal images and time points, brightness and contrast were adjusted linearly and equally. Three-dimensional reconstructions were done with the Imaris FilamentTracer software (Bitplane, Zurich, Switzerland) in automatic detection mode before manual false-coloring and editing to highlight extra- and intra-cerebral vessels.

### gpr124 mutants generation

Two TALENs targeting exons 7 and 16 of gpr124 were designed and cloned using Golden Gate assembly (Cermak et al., 2011) into the pCS2TAL3RR or pCS2TAL3DD expression vectors (Dahlem et al., 2012). The TALEN targeting exon 7 was composed of the following TAL effector domains RVDs: NN HD NI HD NI HD NI NG NG NN NG HD NN NG NG NI NG and HD HD HD NG NG NI NI NI NI NI HD NI NI HD HD NG. The TALEN targeting exon 16 was composed of the following TAL effector domains RVDs: HD HD NI NG NN NG NG NN HD NI NI NG NN NI HD NN NG and NI NG HD HD NG NN NG HD NI NN NI NN NG NN NI NG NN HD HD NG. One-cell stage embryos were injected with 50 pg total TALEN capped messenger RNA synthesized using the mMESSAGE mMACHINE kit (Ambion, Carlsbad, CA). Mutant alleles were identified by high-resolution melt analysis of PCR products generated with the following primers: gpr124-exon 7-Fo: AGGGTCCACTGGAACTGC; gpr124-exon 7-re: CAATGGAAAGGCAGCCTG; gpr124-exon 16-Fo: GCACACTCTCCTGAACACTAG; gpr124-exon 16-Re: TGCCTGGCATACAATTGGG.

## RNA expression constructs and morpholinos

Capped messenger RNA was synthesized using the mMESSAGE mMACHINE kit (Ambion, Carlsbad, CA). The following expression plasmids were generated and used in this study: full-length zebrafish gpr124 and gpr125 were amplified from 48 hpf cDNA and recombined into BamHI-XhoI digested pCS2 using In-fusion cloning (Takara, Mountain View, CA) with the following primers: gpr124-Fo: CTTGTTCTTTTTGCAGGATCCGCCACCATGTCGGGACCCTGCGTC; gpr124-Re: CTATAGTTCTAGA GGCTCGAGTTATACAGTCGTCTCGCTCTTCC; gpr125-Fo: CTTGTTCTTTTTGCAGGATCCGCCACC ATGTCGGTGCTTTGCGTCC; gpr125-Re: CTATAGTTCTAGAGGCTCGAGCTACACAGTAGTTTCAT GCTTCCAC. Full-length zebrafish reck was amplified from 48 hpf cDNA and recombined into BamHI-XbaI digested pCS2 using In-fusion cloning (Takara, Mountain View, CA) with the following primers: reck-Fo: CTTGTTCTTTTTGCAGGATCCACCATGAGCGGGTGTCTCCAGATC; reck-Re: ACGACTC ACTATAGTTCTAGAGTCAGAGGTCAGAGGTCAGGG. gpr124/125 hybrids were generated by recombining two PCR fragments overlapping by 15 bp derived from the above mentioned constructs into BamHI-XhoI digested pCS2 using In-fusion cloning (Takara, Mountain View, CA). In hybrids 1 and 2, the hinge corresponds to amino acid 742/743 of Gpr124 (VLHP$^{742}$/V$^{743}$IYT) and amino acid 739/740 of Gpr125 (PLHP$^{739}$/V$^{740}$IYA). In hybrids 3 and 4, the hinge corresponds to amino acid 1034/1035 of Gpr124 (HHCF$^{1034}$/K$^{1035}$RDL) and amino acid 1029/1030 of Gpr125 (HHCV$^{1029}$/N$^{1030}$RQD). Splice-blocking morpholinos against erbb3b (TGGGCTCGCAACTGGGTGGAAACAA) (Lyons et al., 2005; Budi et al., 2008), sorbs3 (TTTCCGACAGGGAAAGCACATACCC) (Malmquist et al., 2013), reck (CAGGTAGCAGCCGTCACTCACTCTC) (Prendergast et al., 2012) and gpr124 (ACTGATATTGATT TAACTCACCACA) were obtained from Gene Tools (Eugene, OR) and injected at the one-cell stage at 0.5, 4, 1, and 2 ng, respectively, unless otherwise stated. Injection of a standard control morpholino (up to 8 ng) did not affect brain vasculature or DRG formation.

## Transplantations

Host one-cell stage Tg(kdrl:ras-mCherry)$^{s896}$ embryos were injected with the indicated morpholinos. Donor Tg(kdrl:GFP)$^{s843}$ and host embryos were dechorionated with pronase (1 mg/mL) for 5 min at 28°C in 1/3 Ringer solution supplemented with penicillin (50 U/mL)/streptomycin (50 µg/mL) before being incubated in agarose-coated dishes in the same medium. Twenty to 50 cells were removed from donor embryos at mid-blastula stages and transplanted along the blastoderm margin of age-matched host embryos which were subsequently grown at 28.5°C until the indicated stages. The contribution of GFP$^+$ transplanted cells was assessed using a Leica M165 stereomicroscope and EC position within mosaic vessels was determined using confocal microscopy. Contribution of cells of defined genotype to the tip cell position was calculated as a percentage of the total number of mosaic CtAs or ISVs.

## Transgenic endothelial expression

An enhancer from the fli1a gene was used to direct endothelial transgenic expression (Covassin et al., 2006). Expressing cells were visualized by fusing dll4 and gpr124 coding sequences to EGFP by a self-cleaving viral 2A peptide sequence. Transient mosaic endothelial overexpression was obtained by co-injecting 25 pg of Tol2 transposase mRNA and 25 pg of the pTol2-fli1:egfp, pTol2-fli1:gpr124-2A-EGFP, pTol2-fli1:dll4-2A-EGFP, pTol2-fli1:EGFP-2A-gpr124 or pTol2-fli1:EGFP-2A-dll4 constructs.

## Heat-shock and pharmacological treatments

Manually dechorionated embryos were heat-shocked at 26 hpf for 50 min by transferring them into egg water pre-warmed to 38°C or incubated with inhibitors starting at the 16-somite stage until the indicated developmental stage. The following chemical inhibitors were used: LiCl (100 mM), 1-AKP (1-Azakenpaullone; 2.5 µM), SB 216763 (50 µM), BIO (0.5 µM). LiCl was diluted in egg water, other drugs were prepared in 100% DMSO and diluted to the indicated concentration with egg water. As a control, volume-matched DMSO solutions in egg water were used.

## Microarray analysis

RNA was extracted from 20 Tg(kdrl:GFP)$^{s843}$ gpr124 mutants and 20 wild-type siblings, cDNAs were amplified, labeled with Cy3 (gpr124 mutants) or Cy5 (wild-type siblings), and hybridized to the Agilent Zebrafish (V3), 4x44k Gene Expression Microarray.

## Immunofluorescence and proximity ligation assays

FLAG-Gpr124 and HA-Reck were generated by recombining two PCR fragments overlapping by 15 bp derived from the above mentioned constructs into BamHI-XhoI digested pCS2 using In-fusion cloning (Takara, Mountain View, CA). The FLAG peptide was inserted between the amino acid 48/49 of zebrafish Gpr124 and the HA peptide between the amino acid 23/24 of zebrafish Reck. The mouse Fz4-GFP and human FLAG-Dvl2 constructs were obtained from Addgene (Cambridge, MA). HEK 293T cells were transfected with Lipofectamine 2000 (Life Technologies, Carlsbad, CA) 1 day after seeding and were grown in IBIDI imaging chambers for 2 days before fixation for 15 min in 4% paraformaldehyde. Indirect immunofluorescence and proximity ligation assays were performed with the following antibodies: mouse monoclonal anti-FLAG M2 (F1804; Sigma-Aldrich, St. Louis, MO) diluted at 1/2500, purified polyclonal rabbit anti-HA (H6908; Sigma-Aldrich, St. Louis, MO) diluted at 1/250 and Alexa-conjugated secondary antibodies (Molecular Probes, Carlsbad, CA) diluted at 1/5000. PLAs were performed following the manufacturer's instructions (Sigma-Aldrich, St. Louis, MO). DAPI nuclear counterstaining was performed for 2 min at 5 µg/mL.

## Whole-mount in situ hybridization

The dll4 probe was generated with the following primers: dll4-ISH-Fo:GCAGCTTGGCTCACCTTTCTC and dll4-ISH-Re: TAATACGACTCACTATAGGGAGTCCTTTCTCCTGATGCCTGC and T7 was used for transcription and digoxigenin labelling. For whole-mount in situ hybridization, embryos were fixed in 4% paraformaldehyde overnight at 4°C and processed as described previously (Thisse and Thisse, 2008).

## Luciferase assays in STF cells

STF cells were transfected with FuGeneHD (Promega, Madison, WI) 1 day after seeding in a 96-well tray and were harvested 2 days later. Assays were performed in triplicate, and the ratio of activities of firefly luciferase (expressed from a stably integrated reporter with seven tandem Lef/Tcf binding sites) to Renilla luciferase was determined for each well. Wnt signaling components were expressed from a CMV promoter. The following amounts of plasmid DNA were transfected per well. *Figure 4A*: Renilla luciferase (0.5 ng), Lrp5 or Lrp6 (2.5 ng), Wnt7a or Wnt7b (20 ng), Fz4 (20 ng), Gpr124 (20 ng), Reck (20 ng). *Figure 4B*: Renilla luciferase (0.5 ng), GFP (20 ng), Wnt7a (20 ng), Gpr124 (20 ng when held constant [right panel], or the indicated amount [left panel]), Reck (20 ng when held constant [left panel], or the indicated amount [right panel]). *Figure 4C* and *Figure 4—figure supplement 1A*: Renilla luciferase (0.3 ng), GFP (1.5 ng), Lrp5 or Lrp6 (1.5 ng), Wnt7a (3 ng), Gpr124 (0 or 15 ng), Reck (0 or 15 ng), and Frizzled (15 ng). *Figure 4D*: Renilla luciferase (0.5 ng), GFP (20 ng), Gpr124 (0 or 20 ng), Reck (0 or 20 ng), Wnt, Norrin, or vector control (20 ng). The mouse Reck cDNA (GE Dharmacon; clone BC138065, Lafayette, CO) differs from the reference sequence in the NCBI database by a single amino acid substitution (Thr757 in the cDNA instead of Lys757 in the NCBI database); human Reck has Thr at the corresponding location.

## Quantification and statistical analysis

Statistical analysis was performed using GraphPad software. Data presented in bar graphs represent mean ± SD. p-values were calculated by the one-way ANOVA (post hoc Tukey's test) and Student's t test for multiple and single comparisons of normally distributed data (D'Agostino & Pearson omnibus normality

test), respectively and by the Kruskal–Wallis (post hoc Dunn's test) and Mann–Whitney test for multiple and single comparisons of non-normally distributed data, respectively. p-values of tip cell genotype in mosaic vessels (*Figure 7*) were determined by the exact Fisher test. (*p < 0.05; **p < 0.01).

## Acknowledgements

We thank M. Moser, V. Kruys, L. De Grande, and J. Marchal (IBMM-ULB) for help and support and P. Gut for critical reading of the manuscript. We also thank A. Ayala, M. Alva, P. Lopez Pazmino, and M. Sklar for zebrafish husbandry.

## Additional information

### Competing interests

JN: Reviewing editor, *eLife*. The other authors declare that no competing interests exist.

### Funding

| Funder | Grant reference | Author |
| --- | --- | --- |
| Fonds De La Recherche Scientifique—FNRS | F.4543.15 | Benoit Vanhollebeke |
| European Molecular Biology Organization (EMBO) | ALTF 609-2008 | Benoit Vanhollebeke |
| Human Frontier Science Program | LT000334/2009-L | Benoit Vanhollebeke |
| National Eye Institute (NEI) | R01EY081637 | Jeremy Nathans |
| Ellison Medical Foundation | | Jeremy Nathans |
| Howard Hughes Medical Institute (HHMI) | | Jeremy Nathans |
| Deutsche Forschungsgemeinschaft | SFB 834 | Didier YR Stainier |
| National Institutes of Health (NIH) | 5R03 NS074218-02 | Didier YR Stainier |
| David and Lucile Packard Foundation | | Didier YR Stainier |
| Max-Planck-Gesellschaft | | Didier YR Stainier |
| Fondation ULB | | Benoit Vanhollebeke |
| Action de Recherche Concertée de la Fédération Wallonie-Bruxelles | | Benoit Vanhollebeke |
| National Institutes of Health (NIH) | Medical Scientist Training Program | Chris Cho |
| Fonds pour la formation à la Recherche dans l'Industrie et dans l'Agriculture | | Naguissa Bostaille |

The funders had no role in study design, data collection and interpretation, or the decision to submit the work for publication.

### Author contributions

BV, Conceived the study, Designed and performed experiments, Analyzed data, Wrote the paper; OS, Designed and performed experiments, Wrote the paper; NB, CC, YZ, EM, AG, PC, Performed experiments; SF, NM, Contributed unpublished essential reagent, Supervised the work; JN, DYRS, Supervised the work, Analyzed data, Wrote the paper

### Ethics

Animal experimentation: All animal experiments were performed in accordance with the rules of the State of California, USA at the University of California, San Francisco (protocol approval number: AN090555) and the rules of the State of Belgium at the Université Libre de Bruxelles (protocol approval number: CEBEA-IBMM-2012:65).

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
