## [Decision Letter]

Thank you for sending your work entitled “Tip cell specific requirement for an atypical Wnt/β-catenin pathway during CNS angiogenesis” for consideration at *eLife*. Your article has been favorably evaluated by Janet Rossant (Senior editor and Reviewing editor) and two reviewers.

The Reviewing editor and the reviewers discussed their comments before we reached this decision, and the Reviewing editor has assembled the following comments to help you prepare a revised submission.

The reviewers agreed that this is a highly interesting, innovative and well-presented manuscript, highlighting the heterogeneity of the molecular mechanisms that underlie angiogenesis in different vascular beds. It will have broad impact on the field of angiogenesis.

The major points raised that need to be addressed are:

1) The authors nicely cite and describe the importance of Notch signaling for tip-stalk cell formation and the possible role of Gpr124/Reck in this scenario. In the subsection entitled “Wnt/β-catenin signaling in tip cells”, the authors claim that *dll4* knock down and γ-secretase inhibition by DAPT did not rescue the phenotype of Gpr124MO. Considering however the cited paper by Corada et al., conceptually these experiments are not suited to rescue the Gpr124MO phenotype, as Dll4 should be down-regulated in Gpr124MO due to the diminished β-catenin signaling. In this case the over-expression of Dll4, rather than its inhibition should rescue the phenotype. The authors would need to determine the level of Dll4 expression in the Gpr124 and Reck mutants, which would determine whether tip cell specific β-catenin signaling does regulate Dll4 and thereby contribute to the tip cell expression profile and function. Along the same line, the authors should comment on the arterial differentiation of vessels within the Gpr124 and Reck mutants.

2) The authors do not provide direct evidence that Gpr124 and Reck physically interact at the membrane in a Fzd receptor complex. This should be possible to investigate in STF cells by biochemical assays (co-IP, pull-down assays, proximity ligation assay, etc.), at least in the over-expressing situation. Can colocalization be observed by immunofluorescence in the overexpressing cells?

---

## [Author Response]

*1) The authors nicely cite and describe the importance of Notch signaling for tip-stalk cell formation and the possible role of Gpr124/Reck in this scenario. In the subsection entitled “Wnt/β-catenin signaling in tip cells”, the authors claim that* dll4 *knock down and γ-secretase inhibition by DAPT did not rescue the phenotype of Gpr124MO. Considering however the cited paper by Corada et al., conceptually these experiments are not suited to rescue the Gpr124MO phenotype, as Dll4 should be down-regulated in Gpr124MO due to the diminished β-catenin signaling. In this case the over-expression of Dll4, rather than its inhibition should rescue the phenotype. The authors would need to determine the level of Dll4 expression in the Gpr124 and Reck mutants, which would determine whether tip cell specific β-catenin signaling does regulate Dll4 and thereby contribute to the tip cell expression profile and function. Along the same line, the authors should comment on the arterial differentiation of vessels within the Gpr124 and Reck mutants*.

Our revised manuscript now includes additional mechanistic experiments evaluating the potential implication of *dll4* transcriptional dysregulation in the cerebrovascular defects observed in the absence of the Gpr124/Reck-dependent Wnt/β-catenin signaling. In particular, we tested, using an enhancer from the *fli1a* gene, whether mosaic transgenic endothelial expression of *dll4* in *gpr124-* and *reck*-deficient embryos was sufficient to overcome the absence of endothelial Wnt/β-catenin signaling in the PHBCs. To track expressing cells, *dll4* and *gpr124* coding sequences were fused to *EGFP* by a viral 2A peptide sequence and both sequential orientations were tested. While EGFP-positive cells were detected at equal frequencies in different segments of the vasculature, including the PHBCs, with all tested transgenic constructs, only *gpr124*-expressing cells could restore brain angiogenesis by leading new sprouts at tip cell positions (Figure 7—figure supplement 1). *dll4*-expressing cells remained either within the parental PHBCs or contributed to the ventral connections with the basilar artery. In the trunk, EGFP-positive cells could assume tip or stalk cell positions, indicating that transgenic *dll4* expression was globally compatible with angiogenic behaviors. In addition to these functional data, *dll4* expression could be detected by whole-mount in situ hybridization in the PHBCs both in the presence or absence of Gpr124 function, implying that tip cell-specific gene expression and arterial differentiation are not overall disrupted in the absence of Wnt/β-catenin. Whether *dll4* transcript levels are affected at earlier stages or in a refined spatial pattern within the “pre-tip cells” of PHBCs could not be determined due to the low expression levels of the Notch ligand gene in these vessels. The lack of functional rescue after transgenic endothelial *dll4* expression suggests however a role for Wnt/β-catenin during CNS angiogenesis that goes beyond initiating tip and stalk cell fate decisions through Notch ligand transcriptional control.

*2) The authors do not provide direct evidence that Gpr124 and Reck physically interact at the membrane in a Fzd receptor complex. This should be possible to investigate in STF cells by biochemical assays (co-IP, pull-down assays, proximity ligation assay, etc.), at least in the over-expressing situation. Can colocalization be observed by immunofluorescence in the overexpressing cells*?

As requested, we added new data showing that Gpr124, Reck and Frizzled co-localize in cell culture after over-expression (Figure 5 and Figure 5—figure supplement 1). We additionally provide evidence for interaction between Gpr124 and Reck within this compartment through proximity ligation assays. We have not been able to demonstrate physical interactions by co-IP or pull-down assays and this might reflect weak interactions. A relatively weak interaction may not be implausible because in vivo Gpr124, Reck, Fz, and Lrp5/6 are all confined to a 2-dimensional space (the plasma membrane), so that the steady-state level of the complex may be driven by a high encounter frequency (i.e., a high forward rate). That would allow a reasonable concentration of the complex (that can be detected by in situ proximity ligation assays) even with a relatively high dissociation rate. In detergent solution and with a large 3 dimensional volume during a co-IP experiment, the forward rate would be far lower and the concentration of the complex would be correspondingly lower.